# OCATARI: OBJECT-CENTRIC ATARI 2600 REINFORCEMENT LEARNING ENVIRONMENTS

## ABSTRACT

Cognitive science and psychology suggest that object-centric representations of complex scenes are a promising step towards enabling efficient abstract reasoning from low-level perceptual features. Yet, most deep reinforcement learning approaches rely on only pixel-based representations that do not capture the compositional properties of natural scenes. For this, we need environments and datasets that allow us to work and evaluate object-centric approaches. We present OCAtari, a set of environment that provides object-centric state representations of Atari games, the most-used evaluation framework for deep RL approaches. OCAtari also allows for RAM state manipulations of the games to change and create specific or even novel situations. Our source code is available at `https://anonymous.4open.science/r/OCAtari-52B9`.

## 1 INTRODUCTION

In order to solve complex tasks, humans first extract object-centred representations that enable them to draw conclusions while simultaneously blocking out interfering factors (Grill-Spector & Kanwisher, 2005; Tenenbaum et al., 2011; Lake et al., 2017). Deep reinforcement learning (RL) agents cannot provide any object-centric intermediate representations, necessary to check if a suboptimal behavior is caused by misdetections, wrong object identifications, or a reasoning failure. Such representations also permit simpler knowledge transfer between humans and learning agents, or among different tasks, reducing the number of needed samples (Dubey et al., 2018). Object-centricity also permits to use logic to encode the policy, leading to interpretable agents with better generalization capability (Delfosse et al., 2023). Numerous studies on RL research highlight the importance of object-centricity (*cf.* Figure 1), notably in elucidating agent reasoning, prevent misalignment, and potentially correct them (di Langosco et al., 2022). Notably, studies such as Wu et al. (2023); Zhong et al. leverage Language Model-based Learning to realign agents, relying on natural language descriptions of environment goals with great success. This underscores the need to enhance our understanding of RL agents behavior and to ensure their proper alignment with the intended objectives. According to these studies, the object-centric station extraction is a necessary step to achieve it.

Since the introduction of the Arcade Learning Environment (ALE) by Bellemare et al. (2013), Atari 2600 games have become the most common set of environments to test and evaluate RL algorithms, as depicted in Figure 1. As RL methods are challenging to evaluate, compare and reproduce, benchmarks need to encompass a variety of different tasks and challenges to allow for balancing advantages and drawbacks of the different approaches (Henderson et al., 2018; Pineau et al., 2021). ALE games incorporate many RL challenges, such as difficult credit assignment (Skiing), sparse reward (Montezuma's Revenge, Pitfall), and allow for testing approaches with different focuses, such as partial observability (Hausknecht & Stone, 2015), generalization (Farebrother et al., 2018), sample efficiency (Espeholt et al., 2018), environment modeling (Hafner et al., 2021; Schrittwieser et al., 2020), ...etc. Most of the existing object-centric RL frameworks (Kolve et al., 2017; Li et al., 2021) are concerned with robots solving domestic tasks or navigation systems. Object-centric ALE environments would bring all previously discussed benefits and facilitate future comparisons between classic pixel-based approaches and object-centric ones.

Lake et al. (2017) illustrated that deep agents lack the ability to break down the world into multi-step sub-goals, such as acquiring certain objects while avoiding others. In response, they introduced the Frostbite challenge to assess that RL agents integrate such human-like capabilities. Badia et al. (2020) showed that current ALE agents are suboptimal and suggested the use of enhanced representations.

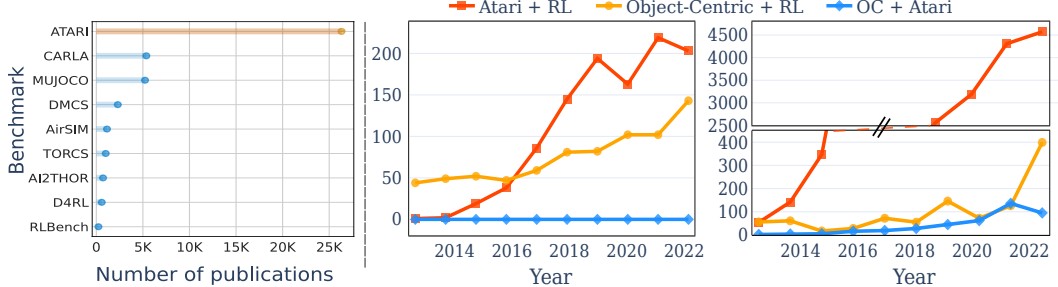

Figure 1: **RL research needs Object-Centric Atari environments.** The Atari Learning Environments (ALE) is, by far, the most used RL benchmark among the ones listed on `paperswithcode.com` (left). Publications using ALE are increasing, together with the number of papers concerned on object-centric RL. As no Object-centric ALE is available yet, the amount of papers on object-centric approaches in Atari is however negligible. Data queried using `dimensions.ai`, based on keyword occurrence in title and abstract (center) or in full text (left and right).

As no such benchmark to test object-centric methods exists yet, we introduce OCAtari, a set of object-centric versions of the ALE games. OCAtari runs the ALE environments while maintaining a list of the objects present in the games, together with their properties. It represents an adequate benchmark for training and comparing upcoming object-centric RL algorithms.

Our framework can be used within object-centric RL algorithms, making it a ressource-efficient replacement for object discovery methods. To train and evaluate these object discovery methods, we also propose the Object-centric Dataset for Atari (ODA), that uses OCAtari to generate a set of Atari frames, together with the properties of the objects present in each game.

Our contributions can be summarized as follows:

- We introduce OCAtari, a RL framework to train and evaluate object-detection and object-centered RL methods on the widely-used Arcade Learning Environment.

- To ease the comparison of object-discovery methods, we introduce ODA, an object-centric dataset, where frames from Atari games are collected using random and DQN agents, together with their object-centric states.

- We evaluate OCAtari and demonstrate that allows to easily generate new challenges for existing Atari-trained method, changing, *e.g.* , object behavior within the games.

We start by introducing the Object-Centric Atari framework. We then experimentally evaluate its detection and speed performances. Before concluding, we touch upon related work.

## 2 THE OBJECT-CENTRIC ATARI ENVIRONMENTS

We here discuss what objects are and how they can be used in RL, then introduce the OCAtari benchmark. Finally, we show how OCAtari helps to generate new versions of the Atari games.

### 2.1 USING OBJECT-CENTRIC DESCRIPTIONS TO LEARN

According to the scientific literature (Thiel, 2011), objects are defined as physical entities that possess properties, attributes, and behaviors that can be observed, measured, and described. Reasoning relies on objects as the fundamental building blocks for our understanding of the world (Rettler & Bailey, 2017). Breaking down the world into objects enables abstraction, generalization, cognitive efficiency, understanding of cause and effect, clear communication, logical inference, and more (see Appendix A for further details). Essentially, objects provide structure, order, and a shared reference point for thinking and communication. They enable us to break down complex situations into manageable components, analyze their interactions, and make informed decisions. The process of reasoning with objects is considered fundamental to human cognition (Spelke et al., 1992; Grill-Spector & Kanwisher, 2005; Tenenbaum et al., 2011; Lake et al., 2017).

However, the concept of an object may vary depending on the situation. Object-centric learning often involves bounding boxes that contain objects and distinguish them from the background (Lin et al., 2020; Delfosse et al., 2022). Static objects, such as the maze in MsPacman or the limiting walls in Pong, are typically considered as part of the background. In this approach, we consider *objects* to be sufficiently small elements, relative to the agent, with which the agents can interact. Excluding "background objects" when learning to play Pong with object-centric inputs is not problematic. However, it can lead to problems when learning to play games like MsPacman. The learning structure should automatically detect and encode game boundaries, but it may have difficulties with the constraints of maze structures. In some settings, it is necessary to provide a background representation. Thus, OCAtari provides both renderings and object-centric descriptions of the states.

## 2.2 THE OCATARI FRAMEWORK

In OCAtari, every object is defined by its category (*e.g.* "Pacman"), position (x and y), size (w and h), and its rgb values. Objects may have additional characteristics such as orientation (*e.g.* the Player in Skiing, *cf.* Figure 4) or value (*e.g.* oxygen bars or scores) if required. Objects vital for gameplay are distinguished from those that are components of the Head-up-Display (HUD) or User Interface (UI) elements (*e.g.* score, number of lives). Generally speaking, the role of HUD objects is to provide additional information about the performance of the playing agent. Although learning agents are capable of ignoring such elements, in our environments a boolean parameter is available to filter out HUD related elements. A list of the considered objects for each game can be found in Appendix F.

To extract objects, OCAtari uses either a Vision Extraction Method (VEM) or a RAM Extraction Method (REM), that are depicted in Figure 2.

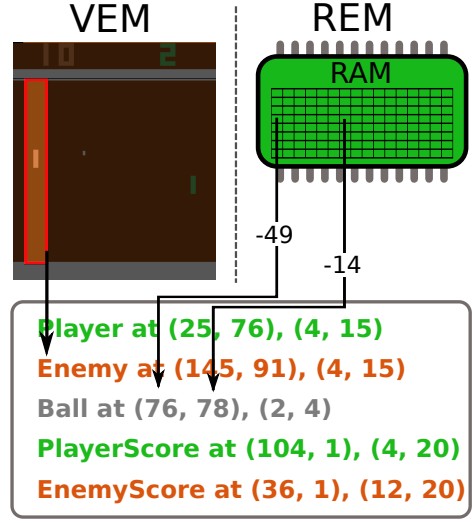

Figure 2: **OCAtari can extract object-centric descriptions using two methods**: the RAM Extraction method (REM) and the Vision Extraction method (VEM).

**VEM: the Vision Extraction Method.** The most straightforward method for extracting objects from Atari game frames involves using simple computer vision techniques. Considering the limited accessible RAM available to Atari developers, most objects are defined by a restricted set of pre-established colors (i.e., RGB values). At each stage, objects are extracted using color-based filtering and an object position priority. Pong consists of three primary objects and two HUD objects, each assigned a single RGB value (*cf.* Figure 2). The opponent always appears above the white threshold, and their paddle is always within the red rectangle. Using this technique, it is possible to accurately extract all present objects. Please note that the detection only detects what is visible in the frame, not objects that are blinking, overlapping, etc.

**REM: the RAM Extraction Method.** ALE provides the state of the emulator's RAM, which contains information about the games' objects. This has led Sygnowski & Michalewski (2016) to use the RAM as states for RL agents. However, much of the non-relevant information is present in the RAM for the policy (*e.g.* time counter, HUD element information). Several games, for example, use bitmaps or encode various information quantities such as object orientation, offset from the anchor, and object category together within one byte. These noisy inputs removes easy interpretation possibilities for these agents behaviors. To address these problems, Anand et al. (2019) have proposed AtariARI, a wrapper around some Atari environments, that provides some the RAM positions, describing where some specific information is encoded. Nonetheless, raw RAM information is not enough. Take, for instance, in *Kangaroo*, the player's position corresponds to various RAM values, that also encode its heights using categorical values. Simply providing some uninfluenced RAM positions does not reflect the object-centric state. Similar to AtariARI, REM extracts the information from the RAM, but processes it to directly provide an interpretable object-centric state, that matches the one of VEM (*cf.* Figure 2). To determine how the game's program processes the RAM information, we task human, random, or DQN agents with playing the games while using VEM to track the depicted objects. We then establish correlations between objects properties (*e.g.* positions) and each of the 128 bytes of the

Table 1: **Games supported by AtariARI and OCAtari.** ✓ describes that all necessary information about the objects are given. ∼ denotes that some necessary information to play the game is lacking. We provide detailed explanation for each of these games in Appendix I. All games missing in this table are neither supported by AtariARI nor OCAtari yet.

| Extraction Method | Alien | Amidar | Assault | Asterix | Asteroids | Atlantis | BattleZone | BeamR. | Berzerk | Bowling | Boxing | Breakout | Carnival | Centipede | ChopperC. | CrazyC. | DemonA. | FishingD. | Freeway | Frostbite | Gopher | Hero | IceHockey | Kangaroo | Montezum. | Ms.Pacman | Pitfall | Pong | PrivateE. | Q*Bert | RiverRaid | RoadR. | Seaquest | Skiing | SpaceInv. | Tennis | Venture | VideoP. | YarsR. | Sum of G. |
|---|---|---|---|---|---|---|---|---|---|---|---|---|---|---|---|---|---|---|---|---|---|---|---|---|---|---|---|---|---|---|---|---|---|---|---|---|---|---|---|---|
| ARI | | | ✓ | | | | | | ✓ | ✓ | ✓ | ✓ | | | | | ∼ | | ✓ | ✓ | | ∼ | ✓ | ✓ | ✓ | ✓ | ✓ | ∼ | ∼ | | | ∼ | | | ∼ | ✓ | ✓ | ✓ | ✓ | 16 (22) |
| REM | ✓ | ✓ | ✓ | ✓ | ✓ | ∼ | | ✓ | ✓ | ✓ | ✓ | ✓ | ✓ | ✓ | ✓ | ✓ | ✓ | ✓ | ✓ | ✓ | ✓ | ✓ | ✓ | ✓ | ✓ | ✓ | ✓ | ✓ | ✓ | ✓ | ✓ | ✓ | ✓ | ✓ | ✓ | ✓ | ✓ | ✓ | ∼ | 36 (38) |
| VEM | ✓ | ✓ | ✓ | ✓ | ✓ | ✓ | ✓ | ✓ | ✓ | ✓ | ✓ | ✓ | ✓ | ✓ | ✓ | ✓ | ✓ | ✓ | ✓ | ✓ | ✓ | ✓ | ✓ | ✓ | ✓ | ✓ | ✓ | ✓ | ✓ | ✓ | ✓ | ✓ | ✓ | ✓ | ✓ | ✓ | ✓ | ✓ | ✓ | 39 |

Atari RAM representation. We can also modify each RAM byte and track the resulting changes in the rendered frames. All these scripts are documented and released along with this paper.

REM, being based on semantic information, is capable of tracking moving objects. Conversely, VEM only furnishes consecutive object-centric descriptions where the lists of objects are independently extracted for each state. REM thus enables tracking of blinking objects and moving instances, as proven useful for RL approaches using tracklets (Agnew & Domingos, 2020; Liu et al., 2021).

**The OCAtari package.** We provide an easy-to-use documented[1] `ocatari` package. OCAtari is designed as a wrapper around the Arcade Learning Environment (ALE) (Bellemare et al., 2013). ALE supports is the primary, most-used framework for testing deep RL methods. To ease its use, ALE was integrated in the OpenAI gymnasium package. To allow an easy swap between ALE and OCAtari environments, we follow the design and naming of the ALE and have reimplemented methods from the ALE framework for OCAtari (*e.g.* `step`, `render`, `seed`, ...). In addition, we added new methods like `get_ram` and `set_ram`, to easily allow RAM manipulation. OCAtari also provide a buffer that contains the last 4 transformed (*i.e.* black and white, $84 \times 84$) frames of the game, as it has become a standard of state representations, notably used in, *e.g.* , DQN (Mnih et al., 2015), DDQN (van Hasselt et al., 2016) and Rainbow (Hessel et al., 2018) algorithms.

As shown in Table 1, our image processing method VEM covers 39 games, while REM covers 38 games at the time of writing. While these already constitute a diverse set of environments, we will continue to add newly supported games in both REM and VEM and complete what we have started. OCAtari is also openly accessible under the MIT license.

**ODA, an Object-centric Dataset for Atari.** OCAtari enables training policies using an object-centric approach to describe RL states for various Atari games. It can serve as a fast and dependable alternative to methods that discover objects. To compare object-centric agents to classic deep ones, it is necessary to train an object detection method and integrate it into the object-centric playing agent, *e.g.* , as shown by Delfosse et al. (2022). To train and compare the object detection methods, we introduce the **O**bject-centric **D**ataset for **A**tari (ODA), a preset selection of frames from the Atari games covered by OCAtari. For each game, ODAs incorporates sequential states, where for each state, the $210 \times 160$ RGB frame is stored with the list of objects found by both VEM and REM procedure (otherwise the game sequence is discarded). The HUD elements are separated from the game objects. Every additional object information contained from the RAM is also saved. As trained agents with varying capabilities can expose different parts of the environment, especially in progressive games where agents must achieve a certain level of mastery to reveal new parts of the game, it is necessary to fix the agents that are used to capture these frames (Delfosse et al., 2021). The frames are extracted using both a random and a trained DQN agent to cover numerous possible states within each game, that should incorporate states encountered by learning agents. In many games, *e.g.* , Montezuma's Revenge or Kangaroo, such agents are not good enough to access every level of the game. However, as the level part is also stored in RAM, we let the agent start in different part of the game by manipulating the RAM. We choose to build our dataset out of 30% of games from the random agent and 70% of the games based on the DQN agent. All needed information, as well as the models used to generate ODA, are provided within the OCAtari repository.

---

[1] https://anonymous.4open.science/r/OCAtari-52B9

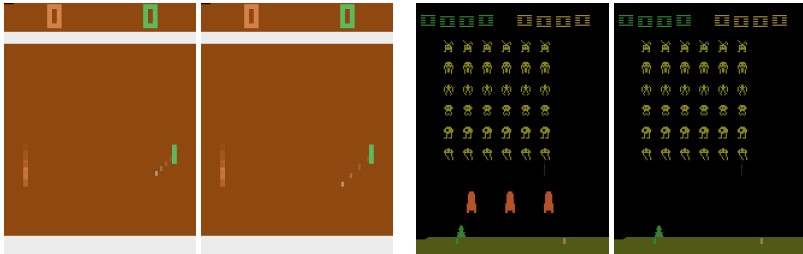

Figure 3: **Manipulating the RAM can introduce novelty into known games such as Pong.** Using the information gathered by the RAM mode, OCAtari is able to manipulate the RAM representation of a state and change game elements or behavior in real time, without recompiling the game. On the left side, we introduce hard-fire Pong, a version where the player can "fire" the ball to increase its velocity. On the right side, we introduce no-shield space invaders.

## 2.3 CREATING NEW CHALLENGES VIA RAM ALTERING.

Modified Atari games are ideal for testing continual reinforcement learning properties with less effort than creating new environments. ALE also allow to alter the RAM values, enabling us to alter the games inner workings. However, modifying game behavior requires a deep understanding of how the RAM information values are processed. OCAtari' REM module makes this process transparent, showing *e.g.* how to derive the objects' positions from anchors and offsets.

This allows us to easily modify games, to *e.g.* reenable multiplayer interfaces and train adversarial policies in games like Pong or Tennis. We also created a modified Pong variant (*cf.* Figure 3, left), in which players can increase the ball speed by pressing the fire button. Similar adaptations introduce unique dynamics, introduce new goals or modify the objects colors. Another example is a more difficult version of SpaceInvaders, without the shields (*cf.* Figure 3, right).

These adaptable environments aid the development and improvement of both pixel-based and OC approaches, addressing challenges like interpretability and robustnes (Delfosse et al., 2023), or generalization (Farebrother et al., 2018). They can also contribute to develop continual RL methods, as done by Tomilin et al. (2023) on Doom.

## 3 EVALUATING OCATARI

In this section, we first present the evaluation of the detection and speed performance of our OCAtari methods. We then explain how REM can be used to train all the parts of object-centric RL agents. Finally, we compare OCAtari to AtariARI.

**Setup**. To evaluate the detection capabilities of REM, we use a random agent (that represents any untrained RL agent), as well as a DQN and, if available, a C51 agent (Bellemare et al., 2017), both obtained from Gogianu et al. (2022)[2]. For reproducibility, every used agent is provided with our along with our codebase. The RL experiments utilized the PPO implementation from stable-baselines3 (Raffin et al., 2021) on a 40 GB DGX A100 server. In each seeded run, 1 critic and 8 actors are utilized per seed over 10M frames. The seeds used were 0, 8, and 16. Training all seeds simultaneously took about 2 hours. Since these experiments do not involve visual representation learning, we utilize an $2 \times 64$ MLP architecture with the hyperbolic tangent as the activation function. This MLP policy architecture is the default for PPO in stable-baselines3. As developing RL agents is not our focus, we did not conduct any fine-tuning or hyperparameter search. Doing so improved the sample efficiency of OC agents in Pong. Further details on these experiments can be found in Appendix D.

## 3.1 EVALUATING OCATARI FOR OBJECT EXTRACTION

**Correctness and Completeness of the Object Extraction.** Atari 2600 games were mainly produced in the 1980s. Due to limitations in memory size, memory access speed, and CPU frequency. Thus, the RAM status is quite different from the object-centric approach that may be used today. Furthermore, certain games, like Riverraid, encode bitmaps over some objects with many information parts (i.e.,

---

[2]https://github.com/floringogianu/atari-agents

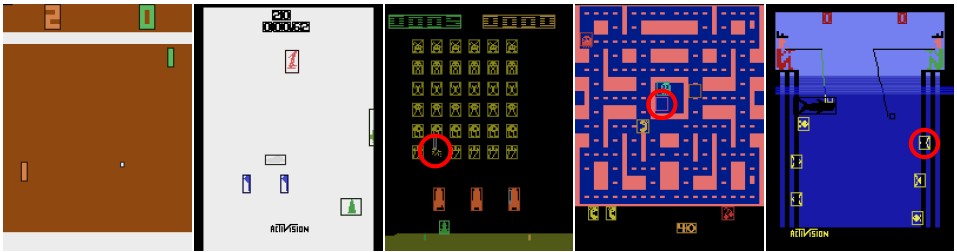

Figure 4: **Qualitative evaluation of OCAtari's REM.** Frames from our OCAtari framework on 5 environments (Pong, Skiing, SpaceInvaders, MsPacman, FishingDerby). Bounding boxes surround the detected objects. REM automatically detects blinking (MsPacman), occluded (FishingDerby) objects, and ignore *e.g.* exploded objects (SpaceInvaders) that vision methods falsely can pick up.

Table 2: **REM reliably detects the objects within the frames of each developed games**. Measuring precision, Recall, F1-Score and IOU of REM (using VEM as baseline) in a diverse set of Atari games. Random and DQN agents were used to capture a large varivety of state spaces. High values being displayed in blue going over green to red for low values. For a more detailed table, *cf.* Appendix F.

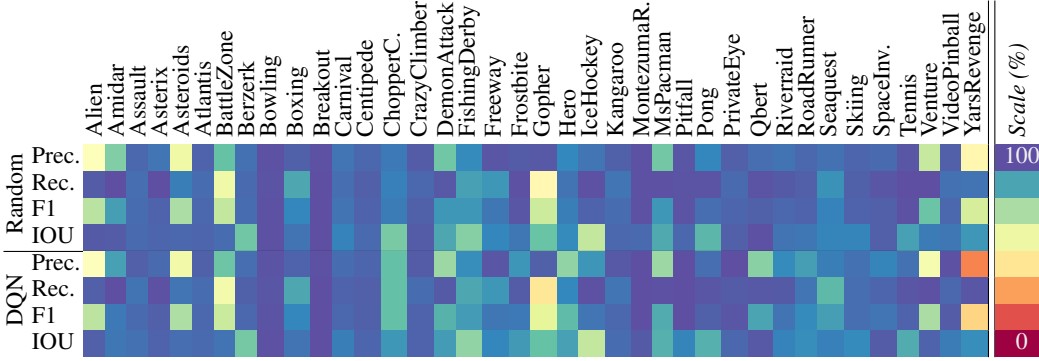

object class, positional offset from an anchor, orientation, staticity) contained within a single byte. It is essential to evaluate whether REM can detect objects accurately and consistently if we aim to use it to train RL agents. We apply the REM of the OCAtari on each supported game for quantitative evaluations, using VEM as the "ground truth", where we used human reviewers to check the correctness and completeness of our output. Frames are sampled using the random and DQN agents. We then compare the object-centric states of both extraction methods (VEM and REM) to check for any differences. For each game, we let the agents play until 500 frames were collected. A prevalent metric for evaluating detection performance is the intersection over union (IOU) (Rezatofighi et al., 2019). However, this metric becomes irrelevant for small objects (such as the ball in *e.g.* Pong, Tennis, or missiles in *e.g.* Atlantis, Space Invaders), as these objects have a size below 4 pixels. Thus, we also calculate precision, recall, and F1-scores for each object category in every game. For these metrics, an object is considered correctly detected if it is within 5 pixels of the center for both detection methods.

In Table 2, we report these metrics, in percentage, averaged over every object category. A perfect score here means that every object-centered state extrated using REM is identical to the VEM ones (*i.e.* that every detected object is obtained with both method for every frame). A lower precision means that some objects detected using REM are not detected by VEM. In MsPacman, the ghost can blink and objects can overlap, which explains why the precision in this game is slightly lower. This can be observed in the per-category tables (*cf.* Appendix F). The mean F1-score, however, is an effective indicator of the overall performance, as it assesses both previously mentioned metrics, using a harmonic mean to find a balance between minimizing false positives and false negatives. In general, the table results indicate that the games included in REM have exceptionally high detection performance. We opted to allow the RAM extraction method to monitor concealed objects, regardless of its effects on the precision of our framework, as it can be used to train object tracking methods that employ tracklets (e.g., Agnew & Domingos 2020) or Kalman filters (e.g., Welch et al. 1995). The

slight differences in ball position and size, which do not impact gameplay, cause the small IOU in Pong and Tennis. Another reason for misdetection is typically due to object occlusion behind the background or that, in many games, the game environment freezes after specific events, such as when a point is scored or when the player dies. In some cases, objects are missing from a few frames, but our RAM extraction approach keeps them listed as objects. Although this decreases the detection scores, it does not affect RL agents since, for these frames, the environment is not interactive.

In Table 3, we compare the detection performances (F-scores) of REM (94%) to two object-centric representation methods (OCR) used on Atari games: SPACE (31%) and SPOC (77%). REM obtains better performances than both. As explained by Delfosse et al. (2022), the detection of such small objects by neural networks remains a challenge, as Atari games entities are composed of few pixels. OCAtari directly does not extract representation, but provides the object classes (from the deterministic RAM information process), that can be used to train the such classifiers.

**Comparing the RAM and Visual Extraction Method.** As explained in the previous section, REM has many advantages over VEM, but relies on an accurate understanding of the game mechanics and their reproduction in the code. REM can track blinking or overlapping objects, and its most significant advantage over VEM is the computational efficiency of the RAM extraction procedure. This is because REM does not needs to process color filtering for each object category, enabling fast and energy-efficient training of object-centric agents. Getting object-centric descriptions of the states using REM is on average 50 times faster than visual object detection (*cf.* Figure 8 in Appendix J). However, REM can also find elements that are not visually represented. This does not have to be a disadvantage, but it is a clear difference to simple visually based approaches not using any tracking. Overall, VEM can be the better choice, if the goal is to extract only visible objects. However, to track objects, REM is a better choice, as it is necessary to distinguish *e.g.* two overlapping blinking objects from a non-blinking entity. In cases where time or costs are important, like in training and using object-centric RL methods, REM should be preferred over VEM.

## 3.2 USING OCATARI AND ODA TO TRAIN OBJECT-CENTRIC RL AGENTS

To fully evaluate that OCAtari allows training object-centric RL agents, we directly trained RL agents using our REM with 3 seeded Proximal Policy Optimization (PPO) agents in 5 different environments. We provide these agents only with the positional information of the objects. Specifically, these correspond to the x and y positions of each object detected in the last two frames. Our trained models are available in our public repository. As depicted in Figure 5, OCAtari allows simple PPO agents to learn to master Atari games, showing that these environments can be used to train object-centric RL agents. Our framework can be used for evaluating more adapted object-centric RL methods.

Overall, we have shown that OCAtari can be used to train or evaluate any part of an object-centric RL agent. ODA can be used to train the object detection methods that extract objects (i.e. position and internal representations or embeddings), but also to train classifiers that extract categories from the extracted embeddings or be used in supervised learning directly. Since REM allows object tracking, it can be used to train or evaluate Kalman filters or tracklets methods. Lastly, OCAtari allows (especially with REM) to be used directly as part of object-centric methods.

## 3.3 OCATARI VS ATARIARI

Before comparing the two frameworks in more detail, it should be noted that both approaches rely on extracting information in RAM. Anand et al. (2019) used commented disassembled source code of various Atari games for their AtariARI framework to find the corresponding RAM location of objects and to annotate the RAM state. In doing so, AtariARI annotated every piece of information found in the code. Unfortunately, providing only the RAM positions is often not enough to get a directly human-interpretable, object-centric description of the state. As shown in Figure 2, even when positions are encoded directly, offsets are applied to objects during the rendering phase. We also found that for some games the information provided by AtariARI is incomplete or insufficient to play the game (*cf.* Table 1 and Appendix I). OCAtari involves intricate computations, such as deriving the x and y positions from anchors and offsets, looking up potential presence indicators (e.g. for objects that have been destroyed), ensuring that our models genuinely acquire object-centric state descriptions. This allows to directly apply human interpretable concepts (*e.g.* `closeby`, `below`, . . . ) and thus use *e.g.* logical reasoning to play the game, as done by Delfosse et al. (2023). OCAtari is

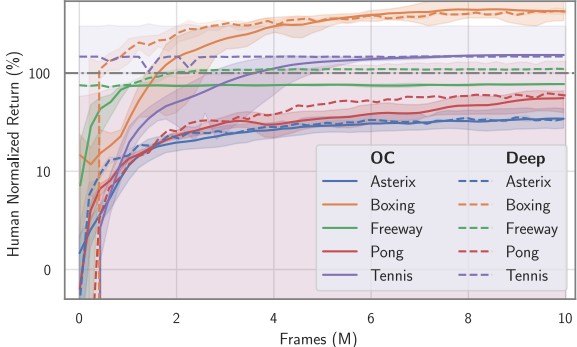

| Game | SPACE | SPOC | REM |
|---|---|---|---|
| Boxing | 24.5 | 70.5 | **90.1** |
| Carnival | 48.6 | 90.6 | **93.7** |
| MsPacm. | 0.4 | **90.5** | 87.4 |
| Pong | 10.7 | 87.4 | **94.3** |
| Riverraid | 45.0 | 76.6 | **95.7** |
| SpaceInv. | 87.5 | 85.2 | **96.9** |
| Tennis | 3.6 | 40.2 | **99.3** |
| Average | 31.5 | 77.3 | **93.9** |

Figure 5: **OCAtari (REM) permits learning of object-centric (OC) RL agents**. The OC PPO agents perform similar to the pixel-based PPO (Deep) agents' and humans on 5 Atari games.

Table 3: **Object detection is still challenging in Atari.** SPACE and SPOC (Delfosse et al., 2022), object detection SOTA, are inferior, in terms of F1 scores.

already covering (19) more games, and we are adapting the rest of the Atari 2600 game collection of ALE. Finally, as explained in our methods section, OCAtari allows the user to easily create variations of the existing games by manipulating the RAM state.

## 4 RELATED WORK

Using Atari as a test bed for deep reinforcement learning has a well-established history. Perhaps best-known here is the work by Mnih et al. (2015), who introduced DQN by testing it on 7 different Atari 2600 games. In the following years, Atari was repeatedly used as a test bed for various approaches, well-known ones being Rainbow DQN or Rainbow Hessel et al. (2018), Dreamer-V2 (Hafner et al., 2020), MuZero (Schrittwieser et al., 2020), Agent57 (Badia et al., 2020) or GDI (Fan et al., 2021). Although RL has achieved fantastic performance in Atari, we still have a lot of challenges left, like the exploration problem (Bellemare et al., 2016; Ecoffet et al., 2019; 2021), efficiency (Kapturowski et al., 2022), planning with sparse or weak rewards (Hafner et al., 2020; Schrittwieser et al., 2020).

Atari is still a challenging and important environment, not only for humans, but also for RL (Toromanoff et al., 2019). Besides classical benchmarks, that deal with the agents' playing strength and try to reach higher and higher scores, we need other benchmarks, *e.g.* , including the efficiency of learning (Machado et al., 2018). In recent years, more benchmarks have been introduced. Toromanoff et al. (2019) and Fan (2021) have both proposed benchmarks for DRL in Atari which add additional metrics to measure performance. Atari 100k, introduced by Kaiser et al. (2020), limits the sample number to test efficiency. Aitchison et al. (2023) have selected small but representative subsets of 5 ALE environments, that will soon be part of OCAtari. Shao et al. (2022) introduced a partial observable Atari benchmark, called Mask Atari, designed to test specifically POMDPs. Anand et al. (2019) introduced AtariARI, an environment for representation learning, which we used as inspiration for this work. A detailed comparison with AtariARI can be found in section 3.3.

Object-centric (or centered) RL involves breaking down raw pixel states into distinct objects and considering their interactions and relationships. This helps to deal with problems like sample inefficiency or missgeneralization (Zambaldi et al., 2019; Mambelli et al., 2022; Stanić et al., 2022).

Following the idea of object-centricity, the usage of object-centered representations (OCR), *e.g.* , Eslami et al. (2016); Kosiorek et al. (2018); Jiang & Luo (2019); Greff et al. (2019); Engelcke et al. (2020); Lin et al. (2020); Locatello et al. (2020); Kipf et al. (2022); Elsayed et al. (2022); Singh et al. (2022a;b) instead of single-vector state representation, used by *e.g.* , Mnih et al. (2015); Hessel et al. (2018) tackle some of the open challenges mentioned above and showing promising results. OCR is diverse, and the evaluations are often specific to their domain and target, like reconstruction loss, segmentation quality or object property prediction (Dittadi et al., 2022; Yoon et al., 2023). Existing object-centric environments are VirtualHome (Puig et al., 2018), AI2-THOR (Kolve et al., 2017) or iGibson 2.0 (Li et al., 2021). While these excel in providing realistic 3D environments conducive to AI research, they introduce high-dimensional observations and emphasizes physical interactions, particularly suitable for robotics-oriented studies. However, ALE's prevalence, relatively low-

dimensional state representation, and diverse task sets make it a preferred choice for benchmarking and comparisons, facilitating the evaluation of both pixel-based and object-centric approaches. A few methods regarding OCR, *e.g.* , AtariARI (Anand et al., 2019), SPACE (Lin et al., 2020) as well as SPOC (Delfosse et al., 2022), have already used Atari games as benchmark.

## 5  DISCUSSION

The main aim is to create an environment for object-centric learning. We primarily evaluate OCAtari's object detection accuracy, with a slight exploration of applications like training simple RL agents and generating challenging environments. Results show how RAM-stored information can create new challenges for existing agents. OCAtari is versatile, suitable for training object-tracking methods and developing new object-centric RL approaches. The provided repository includes scripts for locating and analyzing RAM representation information, offering an information bottleneck in the form of object lists rather than exhaustive details per game. While evaluating performance using the ALE is one of the most recognized benchmarks in RL, these evaluations are not without flaws, as explained by Agarwal et al. (2021). The noisy scores do not linearly reflect the agents' learning ability. These games are also created to be played by humans and thus offer many shortcut learning possibilities. Directly evaluating the representations performance helps to understand and measure the quality of the learned internal representation and minimize other effects within the training, as proposed by Stooke et al. (2021). The object-centricity offered by OCAtari also allows providing extra information to the algorithms, such as an additional reward signal based on the distance between *e.g.* the ball and the player's paddle in Pong, or to add other performance measures (*e.g.* how high has the player going in Kangaroo). This could be an interesting idea of how to use OCAtari for reward shaping.

**Societal and environmental impact.** This work introduces a set of RL environments. While OCAtari can be used for training object-tracking algorithms, there's a potential risk if misapplied. However, its main impact lies in advancing transparent object-centric RL methods. This can enhance understanding of agents' decision-making and reduce misalignment issues, potentially uncovering existing biases in learning algorithms with societal consequences. We do not incorporate and have not found any personal or offensive content in our framework.

**Limitations.** In our work, we have highlighted some advantages of object-centricity and created a framework with OCAtari that manages to extract the objects from a variety of Atari games. However, we are limited in the amount of information we can extract. In most games, there are hardcoded static elements. For instance, the x-position of the paddles in Pong or the maze in MsPacman, such information is usually not found in the RAM representation because it is hard-coded within the game program. As such, we cannot extract this information, or only partially. However, this information being static, it could be learned, but the maze being displayed in MsPacman helps understand that Pacman cannot move through it. An interesting consideration here would be whether a combination of our two modes, object-centricity and vision, can be used to extract not only objects but also important information from the backgrounds. Using this additional information like the position of objects in MsPacman to run A* or similar path finding algorithms could also be an interesting way forward.

## 6  CONCLUSION

Representing scenes in terms of objects and their relations is a crucial human ability that allows one to focus on the essentials when reasoning. While object-centric reinforcement learning and unsupervised detection algorithms are increasingly successful, we lack benchmarks and datasets to evaluate and compare such methods. OCAtari fills this gap and provides an easy-to-use diverse set of environments to test object-centric learning on many Atari games. Overall, we hope our work inspires other researchers to look into object-centric approaches, allowing for more interpretable algorithms that humans can interact with in the future. OCAtari will also permit scientists to create novel challenges among existing Atari games, usable on object-centric, deep or hybrid approaches.

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

## A    DETAILS ON OBJECT PERCEPTION AND ITS ADVANTAGES

As described in our manuscript, decomposing the world in terms of objects incorporates many advantages, some of them are:

**Abstraction and Generalization**
Objects allow us to abstract and generalize information. By categorizing similar objects together, we can create concepts and classifications that help us make sense of a wide variety of individual instances.

**Cognitive Efficiency**
Our brains are more efficient at processing and remembering information when it's organized into meaningful chunks. Objects provide a natural way to group related information, making it easier for us to reason about complex situations.

**Predictive Reasoning**
Objects have properties and behaviors that can be predicted based on their past interactions and characteristics. This predictive reasoning is crucial for making informed decisions and anticipating outcomes.

**Cause and Effect**
Objects play a key role in understanding cause-and-effect relationships. By observing how objects interact and how changes in one object lead to changes in others, we can infer causal connections and predict future outcomes.

**Communication**
Objects provide a shared vocabulary that facilitates communication and understanding. When we refer to objects, we can convey complex ideas more efficiently than describing individual instances or specific situations..

**Logical Inference**
Objects provide a basis for logical reasoning. By identifying relationships between objects, we can deduce logical conclusions and make valid inferences.

# B    DETAILS ON OCATARI

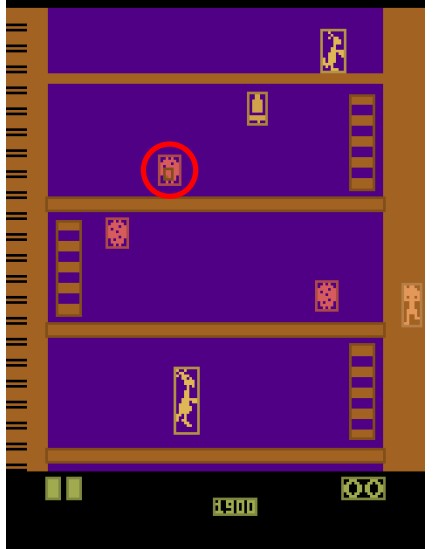

Bell at (93, 36), (6, 11),
Child at (121, 12), (8, 15),
Enemy at (152, 109), (6, 15),
Fruit at (119, 108), (7, 10),
Fruit at (39, 84), (7, 10),
Fruit at (59, 60), (7, 10),
Life at (16, 183), (4, 7),
Life at (24, 183), (4, 7),
Platform at (16, 124), (128, 4),
Platform at (16, 172), (128, 4),
Platform at (16, 76), (128, 4),
Player at (65, 141), (8, 24),
Projectile at (61, 65), (2, 3),
Scale at (132, 132), (8, 35),
Scale at (132, 37), (8, 35),
Scale at (20, 85), (8, 35),
Score at (129, 183), (15, 7),
Time at (80, 191), (15, 5)]

Figure 6: **OCAtari: The object-centric Atari benchmark.** OCAtari maintains a list of existing objects via processing the information from the RAM. Our framework enables training and evaluating object discovery methods and object-centric RL algorithms on the widely used Atari Learning Environments benchmark.

# C    REPRODUCING OUR RESULTS

To reproduce our results, we included the option to run the experiments deterministically. For this purpose, a seed can be specified in the respective scripts. In our experiments, we used the seeds 0 and 42. All supported games can be found in Table 1. Since we are extending the environment permanently, you can also find all supported games in the ReadMe of our repository. To test if a game is supported, you can also use the scripts "test_game" or "test_game_both" depending on if you want to test only one or both modes of OCAtari. Table 2 and all tables in section F are generated by the script "get_metrics". To reproduce and measure the time needed for evaluation, see Figure 8, the script "test_speed" was used. For further information, we recommend checking the documentation of OCAtari under `https://anonymous.4open.science/r/OCAtari-52B9`.

## D  EXPERIMENTAL DETAILS

In our case, all experiments on object extraction and dataset generation were run on a machine with an AMD Ryzen 7 processor, 64GB of RAM, and no dedicated GPU. The dataset generation script takes approximately 3 minutes for one game. We use the same hyperparameters as the Schulman et al. (2017) PPO agents that learned to master the games. Hyperparameter values for Atari environments are derived from the original PPO paper. The same applies to the definitions and values of VF coefficient $c_1$ and entropy coefficient $c_2$. The PPO implementation used and respective MLP hyperparameters are based on stable-baselines3 Raffin et al. (2021). Deep agents have the same hyperparameter values as OCAtari agents but use 'CnnPolicy' in stable-baselines3 for the policy architecture and frame stacking of 4. The Atari environment version used in gymnasium is v4 & v5. This version defines a deterministic skipping of 5 frames per action taken and sets the probability to repeat the last action taken to 0.25. This is aligned with recommended best practices by Machado et al. (2018). We also used the *Deterministic* and *NoFrameskip* features of gymnasium when necessary to make our experiments easier to reproduce. A list of all hyperparameter values used is provided in Table 4.

| | |
|---|---:|
| Actors $N$ | 8 |
| Minibatch size | $32 * 8$ |
| Horizon $T$ | 128 |
| Num. epochs $K$ | 3 |
| Adam stepsize | $2.5 * 10^{-4} * \alpha$ |
| Discount $\gamma$ | 0.99 |
| GAE parameter $\lambda$ | 0.95 |
| Clipping parameter $\epsilon$ | $0.1 * \alpha$ |
| VF coefficient $c_1$ | 1 |
| Entropy coefficient $c_2$ | 0.01 |
| MLP architecture | $2 \times 64$ |
| MLP activation fn. | Tanh |

Table 4: PPO Hyperparameter Values. $\alpha$ linearly increases from 0 to 1 over the course of training.

## E  GENERATING DATASETS

With OCAtari it is possible to create object-centric datasets for all supported games. The dataset consists primarily of a csv file. In addition to a sequential **index**, based on the game number and state number, this file contains the respective image as a list of pixels, called **OBS**. An image in the form of a png file is also stored separately. Furthermore, the csv file contains a list of all HUD elements that could be extracted from the RAM, called **HUD**, as well as a list of all objects that were read from the RAM, called **RAM**. Finally, we provide a list of all elements that could be generated using the vision mode, called **VIS**. An example is given in Table 5.

The generation of the dataset can also be made reproducible by setting a seed. For our tests, we used the seeds 0 and 42. More information at `https://anonymous.4open.science/r/OCAtari-52B9`.

Table 5: An example how an object-centric dataset for Atari looks like after generation.

| Index | OBS | HUD | RAM | VIS |
|---|---|---|---|---|
| 00001_00001 | [[0,0,0]...[255,255,255]] | score at (x,y)(width, height),... | ball at (x,y)(width, height),... | ball at (x,y)(width, height),.... |
| 00001_00002 | [[0,0,0]...[255,255,255]] | score at (x,y)(width, height),... | ball at (x,y)(width, height),... | ball at (x,y)(width, height),.... |
| 00001_00003 | [[0,0,0]...[255,255,255]] | score at (x,y)(width, height),... | ball at (x,y)(width, height),... | ball at (x,y)(width, height),.... |
| ... | | | | |
| 00008_00678 | [[0,0,0]...[255,255,255]] | score at (x,y)(width, height),... | ball at (x,y)(width, height),... | ball at (x,y)(width, height),.... |

# F    DETAILED PER OBJECT CATEGORY RESULTS ON EACH GAME.

In this section, we provide descriptions of each covered game (obtained from `https://gymnasium.farama.org/environments/atari/`) with example frames. For a more detailed documentation, see the game's respective AtariAge manual page[3]. We also share detailed statistics on the object detection capacities of OCAtari for every class of objects detected in each game.

Table 6: A more detailed version of Table 2. Precision, Recall, F1-scores of REM, and intersection over union (IOU) metrics. Frames are obtained using random, DQN and C51 (if available) agents.

| | Random | | | | DQN | | | | C51 | | | |
|---|---|---|---|---|---|---|---|---|---|---|---|---|
| | Prec | Rec | F1 | IOU | Prec | Rec | F1 | IOU | Prec | Rec | F1 | IOU |
| Alien | 51.5 | 97.5 | 67.4 | 97.8 | 50.9 | 97.0 | 66.8 | 97.1 | N/A | N/A | N/A | N/A |
| Amidar | 75.8 | 99.9 | 86.2 | 97.4 | 85.6 | 99.9 | 92.2 | 92.6 | N/A | N/A | N/A | N/A |
| Assault | 95.4 | 94.2 | 94.8 | 95.3 | 97.1 | 93.6 | 95.3 | 93.8 | N/A | N/A | N/A | N/A |
| Asterix | 93.1 | 99.8 | 96.3 | 96.0 | 95.0 | 99.6 | 97.2 | 96.1 | 94.8 | 99.8 | 97.2 | 96.2 |
| Asteroids | 56.6 | 91.6 | 69.9 | 95.4 | 55.4 | 93.3 | 69.5 | 94.4 | N/A | N/A | N/A | N/A |
| Atlantis | 96.3 | 94.6 | 95.5 | 95.8 | 96.6 | 94.7 | 95.7 | 95.0 | N/A | N/A | N/A | N/A |
| BattleZone | 79.9 | 56.2 | 66.0 | 94.7 | 78.7 | 54.7 | 64.5 | 93.7 | N/A | N/A | N/A | N/A |
| Berzerk | 94.1 | 95.2 | 94.6 | 78.4 | 94.3 | 96.5 | 95.4 | 77.4 | N/A | N/A | N/A | N/A |
| Bowling | 99.5 | 99.2 | 99.3 | 99.6 | 99.2 | 98.8 | 99.0 | 99.4 | 99.4 | 99.1 | 99.3 | 99.5 |
| Boxing | 96.5 | 84.5 | 90.1 | 93.5 | 96.1 | 84.5 | 89.9 | 93.4 | 96.8 | 85.6 | 90.9 | 94.1 |
| Breakout | 99.5 | 100.0 | 99.7 | 100.0 | 99.5 | 100.0 | 99.7 | 100.0 | 100.0 | 100.0 | 100.0 | 100.0 |
| Carnival | 93.2 | 94.2 | 93.7 | 90.7 | 94.6 | 96.4 | 95.5 | 91.5 | N/A | N/A | N/A | N/A |
| Centipede | 95.7 | 97.0 | 96.3 | 95.1 | 95.9 | 97.2 | 96.6 | 96.0 | N/A | N/A | N/A | N/A |
| ChopperCommand | 92.2 | 91.4 | 91.8 | 77.2 | 80.3 | 80.2 | 80.2 | 87.3 | 77.9 | 78.5 | 78.2 | 92.0 |
| CrazyClimber | 97.6 | 95.5 | 96.6 | 97.4 | 97.6 | 94.8 | 96.2 | 96.1 | N/A | N/A | N/A | N/A |
| DemonAttack | 78.4 | 98.1 | 87.2 | 84.6 | 71.9 | 97.7 | 82.8 | 86.5 | N/A | N/A | N/A | N/A |
| FishingDerby | 89.2 | 85.6 | 87.3 | 75.2 | 88.8 | 84.6 | 86.6 | 73.6 | 83.2 | 77.9 | 80.5 | 75.7 |
| Freeway | 98.7 | 87.3 | 92.6 | 90.2 | 98.6 | 87.3 | 92.6 | 90.2 | 96.5 | 87.2 | 91.6 | 87.9 |
| Frostbite | 97.6 | 99.5 | 98.6 | 92.7 | 87.5 | 97.5 | 92.2 | 87.1 | 85.5 | 97.1 | 90.9 | 85.4 |
| Gopher | 98.4 | 47.5 | 64.1 | 79.3 | 96.5 | 42.5 | 59.0 | 79.8 | N/A | N/A | N/A | N/A |
| Hero | 89.8 | 93.6 | 91.7 | 89.4 | 73.7 | 89.8 | 81.0 | 85.9 | 80.2 | 93.3 | 86.2 | 87.3 |
| IceHockey | 93.2 | 99.6 | 96.3 | 65.9 | 87.7 | 99.4 | 93.2 | 66.0 | N/A | N/A | N/A | N/A |
| Kangaroo | 96.7 | 93.1 | 94.9 | 95.6 | 98.3 | 93.2 | 95.7 | 94.8 | 96.1 | 93.1 | 94.6 | 95.2 |
| MontezumaRevenge | 99.5 | 99.4 | 99.5 | 95.2 | 100.0 | 100.0 | 100.0 | 97.9 | 100.0 | 100.0 | 100.0 | 98.2 |
| MsPacman | 77.9 | 99.4 | 87.4 | 84.2 | 72.1 | 99.3 | 83.6 | 83.1 | N/A | N/A | N/A | N/A |
| Pitfall | 98.6 | 99.1 | 98.9 | 95.6 | 99.8 | 100.0 | 99.9 | 90.6 | N/A | N/A | N/A | N/A |
| Pong | 90.0 | 99.1 | 94.3 | 81.7 | 94.3 | 98.8 | 96.5 | 83.2 | 93.8 | 97.4 | 95.6 | 84.7 |
| PrivateEye | 97.0 | 95.1 | 96.0 | 97.1 | 99.7 | 97.6 | 98.6 | 95.6 | N/A | N/A | N/A | N/A |
| Qbert | 94.4 | 99.0 | 96.6 | 99.6 | 74.7 | 98.3 | 84.9 | 98.4 | 77.3 | 98.4 | 86.6 | 98.5 |
| Riverraid | 93.5 | 98.0 | 95.7 | 93.6 | 89.3 | 98.0 | 93.5 | 91.0 | N/A | N/A | N/A | N/A |
| RoadRunner | 95.5 | 97.2 | 96.3 | 92.7 | 86.0 | 93.1 | 89.4 | 88.8 | N/A | N/A | N/A | N/A |
| Seaquest | 94.1 | 87.9 | 90.9 | 90.3 | 91.5 | 81.3 | 86.1 | 91.4 | 92.1 | 82.6 | 87.1 | 90.6 |
| Skiing | 95.8 | 96.5 | 96.2 | 90.4 | 94.1 | 94.2 | 94.2 | 89.3 | N/A | N/A | N/A | N/A |
| SpaceInvaders | 95.2 | 98.7 | 96.9 | 97.1 | 90.6 | 95.9 | 93.1 | 97.3 | N/A | N/A | N/A | N/A |
| Tennis | 98.7 | 99.9 | 99.3 | 85.7 | 93.9 | 98.7 | 96.2 | 83.9 | N/A | N/A | N/A | N/A |
| Venture | 65.4 | 99.8 | 79.0 | 92.1 | 54.0 | 99.4 | 70.0 | 96.1 | N/A | N/A | N/A | N/A |
| VideoPinball | 97.1 | 93.9 | 95.5 | 92.9 | 99.5 | 95.7 | 97.5 | 91.8 | N/A | N/A | N/A | N/A |
| YarsRevenge | 47.0 | 93.6 | 62.6 | 87.2 | 23.5 | 98.5 | 37.9 | 88.9 | N/A | N/A | N/A | N/A |
| Mean | 89.4 | 93.5 | 90.4 | 90.9 | 86.4 | 92.7 | 87.8 | 90.4 | 91.0 | 92.1 | 91.3 | 91.8 |

## F.1    ALIEN DETAILS

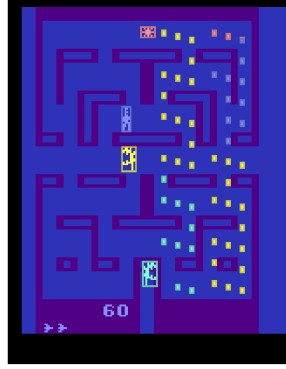

You are stuck in a maze-like space ship with three aliens. You goal is to destroy their eggs that are scattered all over the ship while simultaneously avoiding the aliens (they are trying to kill you). You have a flamethrower that can help you turn them away in tricky situations. Moreover, you can occasionally collect a power-up (pulsar) that gives you the temporary ability to kill aliens.

---

[3]`https://atariage.com/system_items.php?SystemID=2600&itemTypeID=MANUAL`

Table 7: Per class IOU on Alien

|        | Random |      |       |      | DQN  |      |       |      | C51 |     |       |     |
|--------|--------|------|-------|------|------|------|-------|------|-----|-----|-------|-----|
|        | Pr     | Rec  | F-sc  | IOU  | Pr   | Rec  | F-sc  | IOU  | Pr  | Rec | F-sc  | IOU |
| Score  | 100    | 93.8 | 96.8  | 97.6 | 98.0 | 85.4 | 91.2  | 94.7 | nan | nan | nan   | nan |
| Egg    | 49.6   | 97.9 | 65.8  | 98.9 | 48.8 | 97.7 | 65.1  | 98.6 | nan | nan | nan   | nan |
| Life   | 99.8   | 99.6 | 99.7  | 100  | 100  | 100  | 100   | 100  | nan | nan | nan   | nan |
| Pulsar | 67.2   | 82.0 | 73.8  | 80.6 | 66.4 | 82.4 | 73.5  | 79.1 | nan | nan | nan   | nan |
| Player | 83.2   | 99.8 | 90.7  | 58.1 | 80.8 | 99.8 | 89.3  | 58.8 | nan | nan | nan   | nan |
| Alien  | 73.0   | 92.1 | 81.5  | 94.2 | 68.6 | 92.5 | 78.8  | 94.3 | nan | nan | nan   | nan |

## F.2  AMIDAR DETAILS

This game is similar to Pac-Man: You are trying to visit all places on a 2-dimensional grid while simultaneously avoiding your enemies. You can turn the tables at one point in the game: Your enemies turn into chickens and you can catch them.

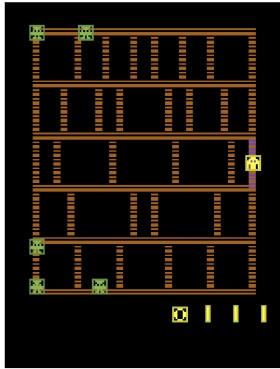

Table 8: Per class IOU on Amidar

|               | Random |      |       |      | DQN  |      |       |      | C51 |     |       |     |
|---------------|--------|------|-------|------|------|------|-------|------|-----|-----|-------|-----|
|               | Pr     | Rec  | F-sc  | IOU  | Pr   | Rec  | F-sc  | IOU  | Pr  | Rec | F-sc  | IOU |
| Life          | 100    | 100  | 100   | 100  | 100  | 100  | 100   | 100  | nan | nan | nan   | nan |
| Monster_green | 60.9   | 99.9 | 75.7  | 95.4 | 75.7 | 99.8 | 86.1  | 87.4 | nan | nan | nan   | nan |
| Score         | 100    | 100  | 100   | 100  | 100  | 100  | 100   | 95.4 | nan | nan | nan   | nan |
| Player        | 97.4   | 100  | 98.7  | 95.5 | 95.6 | 100  | 97.8  | 92.9 | nan | nan | nan   | nan |

## F.3  ASSAULT DETAILS

You control a vehicle that can move sideways. A big mother ship circles overhead and continually deploys smaller drones. You must destroy these enemies and dodge their attacks.

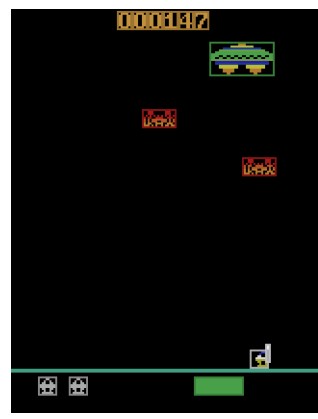

Table 9: Per class statistics on Assault

|  | Random | | | | DQN | | | | C51 | | | |
|---|---|---|---|---|---|---|---|---|---|---|---|---|
|  | Pr | Rec | F-sc | IOU | Pr | Rec | F-sc | IOU | Pr | Rec | F-sc | IOU |
| PlayerScore | 100 | 100 | 100 | 100 | 100 | 100 | 100 | 99.9 | nan | nan | nan | nan |
| MotherShip | 99.8 | 99.8 | 99.8 | 88.7 | 100 | 100 | 100 | 88.6 | nan | nan | nan | nan |
| Lives | 100 | 100 | 100 | 100 | 100 | 100 | 100 | 100 | nan | nan | nan | nan |
| Health | 99.6 | 99.6 | 99.6 | 99.5 | 100 | 100 | 100 | 99.6 | nan | nan | nan | nan |
| Player | 91.8 | 100 | 95.7 | 88.6 | 95.8 | 100 | 97.9 | 81.6 | nan | nan | nan | nan |
| Enemy | 98.4 | 87.6 | 92.7 | 87.1 | 89.0 | 70.5 | 78.7 | 78.5 | nan | nan | nan | nan |
| PlayerMissileHorizontal | 29.0 | 29.1 | 29.1 | 28.5 | 26.7 | 25.5 | 26.1 | 30.2 | nan | nan | nan | nan |
| PlayerMissileVertical | 95.7 | 95.2 | 95.5 | 86.7 | 91.4 | 88.0 | 89.7 | 83.0 | nan | nan | nan | nan |
| EnemyMissile | 20.0 | 21.7 | 20.8 | 69.6 | 44.4 | 38.1 | 41.0 | 67.9 | nan | nan | nan | nan |

## F.4 ASTERIX DETAILS

You are Asterix and can move horizontally (continuously) and vertically (discretely). Objects move horizontally across the screen: lyres and other (more useful) objects. Your goal is to guide Asterix in such a way as to avoid lyres and collect as many other objects as possible. You score points by collecting objects and lose a life whenever you collect a lyre. You have three lives available at the beginning. If you score sufficiently many points, you will be awarded additional points.

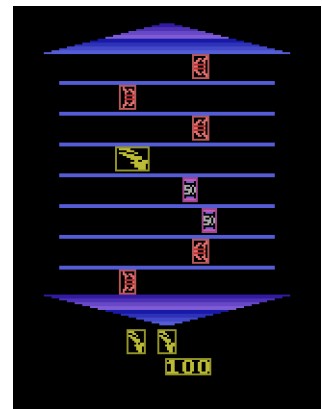

Table 10: Per class statistics on Asterix

|  | Random | | | | DQN | | | | C51 | | | |
|---|---|---|---|---|---|---|---|---|---|---|---|---|
|  | Pr | Rec | F-sc | IOU | Pr | Rec | F-sc | IOU | Pr | Rec | F-sc | IOU |
| Lives | 100 | 100 | 100 | 91.7 | 100 | 100 | 100 | 91.7 | 100 | 100 | 100 | 91.7 |
| Player | 98.4 | 98.4 | 98.4 | 97.6 | 99.6 | 99.6 | 99.6 | 98.8 | 98.6 | 98.6 | 98.6 | 98.5 |
| Score | 100 | 100 | 100 | 100 | 100 | 96.3 | 98.1 | 99.0 | 100 | 99.2 | 99.6 | 99.8 |
| Cauldron | 93.9 | 100 | 96.8 | 99.9 | 96.6 | 100 | 98.3 | 100 | 98.5 | 100 | 99.2 | 100 |
| Reward50 | 90.7 | 100 | 95.1 | 100 | 98.2 | 100 | 99.1 | 100 | 90.6 | 100 | 95.1 | 99.6 |
| Enemy | 85.8 | 100 | 92.3 | 88.0 | 85.4 | 100 | 92.1 | 89.2 | 89.7 | 100 | 94.6 | 89.6 |

## F.5  ASTEROIDS DETAILS

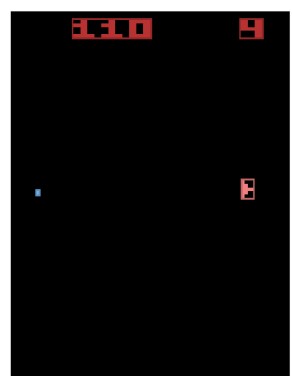

This is a well-known arcade game: You control a spaceship in an asteroid field and must break up asteroids by shooting them. Once all asteroids are destroyed, you enter a new level and new asteroids will appear. You will occasionally be attacked by a flying saucer.

Table 11: Per class IOU on Asteroids

|  | Random | | | | DQN | | | | C51 | | | |
|  | Pr | Rec | F-sc | IOU | Pr | Rec | F-sc | IOU | Pr | Rec | F-sc | IOU |
|---|---|---|---|---|---|---|---|---|---|---|---|---|
| Asteroid | 42.5 | 86.7 | 57.1 | 90.5 | 39.5 | 88.8 | 54.7 | 89.3 | nan | nan | nan | nan |
| Player | 44.3 | 100 | 61.4 | 75.1 | 50.5 | 97.3 | 66.5 | 73.1 | nan | nan | nan | nan |
| Lives | 100 | 100 | 100 | 100 | 100 | 100 | 100 | 100 | nan | nan | nan | nan |
| PlayerScore | 96.0 | 92.5 | 94.2 | 98.5 | 97.8 | 96.1 | 96.9 | 99.1 | nan | nan | nan | nan |
| PlayerMissile | 44.7 | 97.8 | 61.4 | 98.7 | 52.7 | 92.8 | 67.2 | 100 | nan | nan | nan | nan |

## F.6  ATLANTIS DETAILS

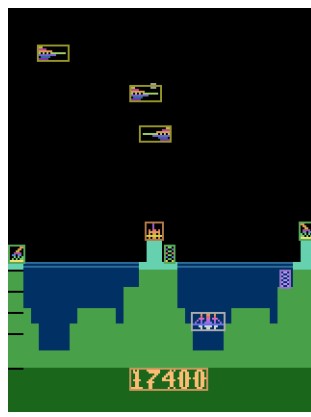

Your job is to defend the submerged city of Atlantis. Your enemies slowly descend towards the city and you must destroy them before they reach striking distance. To this end, you control three defense posts. You lose if your enemies manage to destroy all seven of Atlantis' installations. You may rebuild installations after you have fought of a wave of enemies and scored a sufficient number of points.

Table 12: Per class statistics on Atlantis

| | Random | | | | DQN | | | | C51 | | | |
|---|---|---|---|---|---|---|---|---|---|---|---|---|
| | Pr | Rec | F-sc | IOU | Pr | Rec | F-sc | IOU | Pr | Rec | F-sc | IOU |
| BridgedBazaar | 100 | 99.7 | 99.9 | 99.9 | 100 | 99.7 | 99.8 | 99.9 | nan | nan | nan | nan |
| AcropolisCommandPost | 100 | 96.2 | 98.0 | 99.9 | 100 | 97.8 | 98.9 | 99.8 | nan | nan | nan | nan |
| Sentry | 100 | 100 | 100 | 99.4 | 100 | 100 | 100 | 99.6 | nan | nan | nan | nan |
| AquaPlane | 100 | 97.8 | 98.9 | 100 | 100 | 99.0 | 99.5 | 99.9 | nan | nan | nan | nan |
| Generator | 100 | 99.7 | 99.9 | 87.9 | 99.8 | 99.5 | 99.7 | 85.7 | nan | nan | nan | nan |
| DomedPalace | 100 | 99.5 | 99.8 | 100 | 100 | 99.8 | 99.9 | 99.9 | nan | nan | nan | nan |
| Projectile | 85.1 | 77.4 | 81.1 | 99.9 | 86.7 | 76.2 | 81.1 | 99.2 | nan | nan | nan | nan |
| GorgonShip | 88.6 | 83.0 | 85.7 | 93.1 | 85.7 | 76.0 | 80.6 | 89.9 | nan | nan | nan | nan |
| Score | 100 | 100 | 100 | 99.8 | 100 | 100 | 100 | 100 | nan | nan | nan | nan |
| BanditBomber | 81.0 | 78.2 | 79.5 | 88.2 | 76.3 | 85.3 | 80.6 | 79.4 | nan | nan | nan | nan |

## F.7 BATTLEZONE DETAILS

You control a tank and must destroy enemy vehicles. This game is played in a first-person perspective and creates a 3D illusion. A radar screen shows enemies around you. You start with 5 lives and gain up to 2 extra lives if you reach a sufficient score.

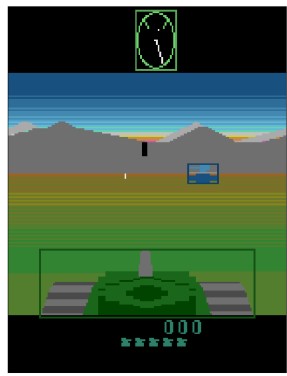

Table 13: Per class IOU on BattleZone

| | Random | | | | DQN | | | | C51 | | | |
|---|---|---|---|---|---|---|---|---|---|---|---|---|
| | Pr | Rec | F-sc | IOU | Pr | Rec | F-sc | IOU | Pr | Rec | F-sc | IOU |
| Radar | 92.0 | 100 | 95.8 | 100 | 92.8 | 94.5 | 93.6 | 99.6 | nan | nan | nan | nan |
| Player | 92.0 | 100 | 95.8 | 100 | 92.8 | 100 | 96.3 | 100 | nan | nan | nan | nan |
| Crosshair | 92.0 | 68.0 | 78.2 | 97.5 | 88.6 | 70.5 | 78.5 | 97.3 | nan | nan | nan | nan |
| Blue_Tank | 27.1 | 50.8 | 35.4 | 57.6 | 27.3 | 47.6 | 34.7 | 54.9 | nan | nan | nan | nan |

## F.8 BERZERK DETAILS

You are stuck in a maze with evil robots. You must destroy them and avoid touching the walls of the maze, as this will kill you. You may be awarded extra lives after scoring a sufficient number of points, depending on the game mode. You may also be chased by an undefeatable enemy, Evil Otto, that you must avoid. Evil Otto does not appear in the default mode.

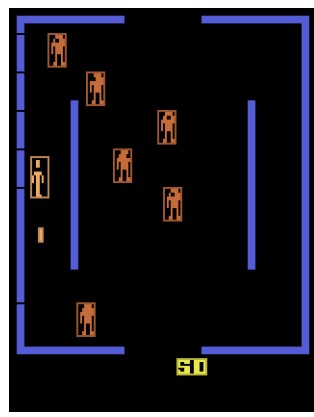

Table 14: Per class statistics on Berzerk

| | Random | | | | DQN | | | | C51 | | | |
| | Pr | Rec | F-sc | IOU | Pr | Rec | F-sc | IOU | Pr | Rec | F-sc | IOU |
|---|---|---|---|---|---|---|---|---|---|---|---|---|
| Logo | 100 | 99.3 | 99.6 | 100 | 100 | 100 | 100 | 100 | nan | nan | nan | nan |
| PlayerMissile | 75.0 | 98.4 | 85.1 | 74.5 | 79.2 | 88.1 | 83.4 | 81.1 | nan | nan | nan | nan |
| Enemy | 97.3 | 98.7 | 98.0 | 77.0 | 98.4 | 100 | 99.2 | 77.3 | nan | nan | nan | nan |
| Player | 90.4 | 98.7 | 94.4 | 66.1 | 97.4 | 99.2 | 98.3 | 54.5 | nan | nan | nan | nan |
| PlayerScore | 95.0 | 73.2 | 82.7 | 85.1 | 98.8 | 91.9 | 95.2 | 96.6 | nan | nan | nan | nan |
| EnemyMissile | 77.4 | 90.0 | 83.2 | 78.8 | 73.8 | 85.7 | 79.3 | 79.2 | nan | nan | nan | nan |
| RoomCleared | 96.3 | 100 | 98.1 | 100 | 98.4 | 100 | 99.2 | 100 | nan | nan | nan | nan |

## F.9 BOWLING DETAILS

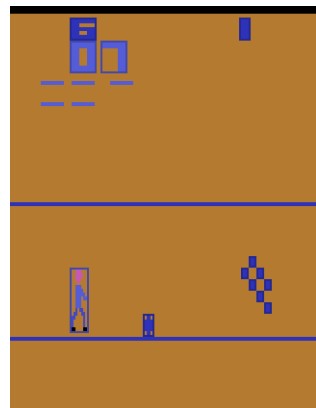

Your goal is to score as many points as possible in the game of Bowling. A game consists of 10 frames and you have two tries per frame. Knocking down all pins on the first try is called a "strike". Knocking down all pins on the second roll is called a "spar". Otherwise, the frame is called "open".

Table 15: Per class statistics on Bowling

| | Random | | | | DQN | | | | C51 | | | |
| | Pr | Rec | F-sc | IOU | Pr | Rec | F-sc | IOU | Pr | Rec | F-sc | IOU |
|---|---|---|---|---|---|---|---|---|---|---|---|---|
| Pin | 99.4 | 100 | 99.7 | 100 | 98.7 | 100 | 99.3 | 100 | 99.4 | 100 | 99.7 | 100 |
| Player | 98.6 | 97.6 | 98.1 | 98.3 | 99.6 | 99.2 | 99.4 | 99.2 | 98.2 | 96.8 | 97.5 | 98.0 |
| PlayerScore | 100 | 100 | 100 | 100 | 100 | 100 | 100 | 100 | 100 | 100 | 100 | 100 |
| Player2Round | 100 | 100 | 100 | 100 | 100 | 100 | 100 | 100 | 100 | 100 | 100 | 100 |
| Ball | 99.0 | 98.8 | 98.9 | 99.6 | 98.4 | 96.9 | 97.6 | 99.1 | 99.0 | 98.6 | 98.8 | 99.3 |
| PlayerRound | 100 | 92.4 | 96.1 | 96.7 | 100 | 89.6 | 94.5 | 95.4 | 100 | 92.1 | 95.9 | 96.6 |

## F.10 BOXING DETAILS

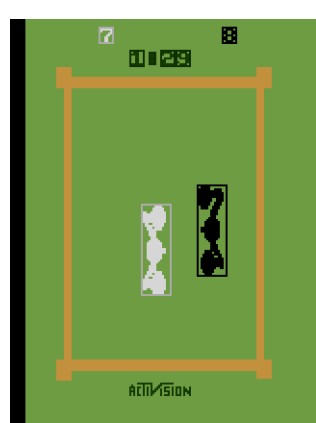

You fight an opponent in a boxing ring. You score points for hitting the opponent. If you score 100 points, your opponent is knocked out.

Table 16: Per class statistics on Boxing

| | Random | | | | DQN | | | | C51 | | | |
|---|---|---|---|---|---|---|---|---|---|---|---|---|
| | Pr | Rec | F-sc | IOU | Pr | Rec | F-sc | IOU | Pr | Rec | F-sc | IOU |
| Enemy | 83.4 | 81.1 | 82.2 | 78.4 | 79.8 | 74.6 | 77.1 | 73.3 | 88.6 | 85.5 | 87.0 | 79.5 |
| PlayerScore | 100 | 82.5 | 90.4 | 87.9 | 100 | 92.8 | 96.2 | 95.5 | 100 | 93.3 | 96.5 | 95.9 |
| Player | 81.6 | 81.6 | 81.6 | 79.3 | 80.8 | 80.8 | 80.8 | 78.4 | 79.6 | 79.1 | 79.4 | 76.7 |
| EnemyScore | 100 | 45.9 | 62.9 | 89.8 | 100 | 44.9 | 62.0 | 87.1 | 100 | 45.5 | 62.5 | 88.7 |
| Logo | 100 | 100 | 100 | 100 | 100 | 100 | 100 | 100 | 100 | 100 | 100 | 100 |
| Clock | 100 | 100 | 100 | 100 | 100 | 100 | 100 | 100 | 100 | 100 | 100 | 100 |

## F.11 BREAKOUT DETAILS

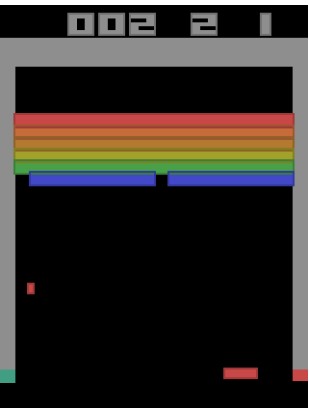

Another famous Atari game. The dynamics are similar to pong: You move a paddle and hit the ball in a brick wall at the top of the screen. Your goal is to destroy the brick wall. You can try to break through the wall and let the ball wreak havoc on the other side, all on its own! You have five lives.

Table 17: Per class statistics on Breakout

| | Random | | | | DQN | | | | C51 | | | |
|---|---|---|---|---|---|---|---|---|---|---|---|---|
| | Pr | Rec | F-sc | IOU | Pr | Rec | F-sc | IOU | Pr | Rec | F-sc | IOU |
| Player | 98.8 | 100 | 99.4 | 100 | 97.4 | 100 | 98.7 | 100 | 99.8 | 100 | 99.9 | 100 |
| BlockRow | 100 | 100 | 100 | 100 | 100 | 100 | 100 | 100 | 100 | 100 | 100 | 100 |
| Live | 100 | 100 | 100 | 99.9 | 100 | 100 | 100 | 99.9 | 100 | 100 | 100 | 100 |
| PlayerScore | 100 | 100 | 100 | 100 | 100 | 100 | 100 | 100 | 100 | 100 | 100 | 100 |
| PlayerNumber | 100 | 100 | 100 | 100 | 100 | 100 | 100 | 100 | 100 | 100 | 100 | 100 |
| Ball | 93.1 | 100 | 96.4 | 100 | 93.3 | 100 | 96.5 | 100 | 90.5 | 100 | 95.0 | 100 |

## F.12 CARNIVAL DETAILS

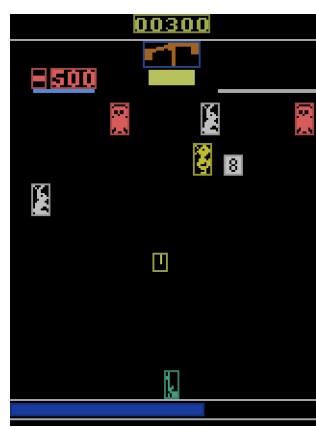

This is a "shoot 'em up" game. Targets move horizontally across the screen and you must shoot them. You are in control of a gun that can be moved horizontally. The supply of ammunition is limited and chickens may steal some bullets from you if you don't hit them in time.

Table 18: Per class statistics on Carnival

| | Random | | | | DQN | | | | C51 | | | |
| | Pr | Rec | F-sc | IOU | Pr | Rec | F-sc | IOU | Pr | Rec | F-sc | IOU |
|---|---|---|---|---|---|---|---|---|---|---|---|---|
| Duck | 98.8 | 93.7 | 96.2 | 95.5 | 98.3 | 93.4 | 95.8 | 94.4 | nan | nan | nan | nan |
| PlayerScore | 86.6 | 76.4 | 81.2 | 100 | 98.4 | 96.9 | 97.6 | 100 | nan | nan | nan | nan |
| Wheel | 100 | 100 | 100 | 89.8 | 100 | 98.9 | 99.5 | 88.5 | nan | nan | nan | nan |
| FlyingDuck | 40.4 | 91.3 | 56.0 | 82.2 | 42.9 | 82.9 | 56.5 | 82.9 | nan | nan | nan | nan |
| Owl | 99.0 | 97.7 | 98.3 | 97.6 | 98.9 | 96.2 | 97.5 | 95.9 | nan | nan | nan | nan |
| ExtraBullets | 98.1 | 84.9 | 91.0 | 97.8 | 98.1 | 90.2 | 94.0 | 96.6 | nan | nan | nan | nan |
| Player | 100 | 100 | 100 | 100 | 100 | 100 | 100 | 100 | nan | nan | nan | nan |
| Rabbit | 97.0 | 95.8 | 96.4 | 98.0 | 95.3 | 97.3 | 96.3 | 97.4 | nan | nan | nan | nan |
| AmmoBar | 90.0 | 100 | 94.7 | 100 | 98.4 | 99.8 | 99.1 | 99.8 | nan | nan | nan | nan |
| PlayerMissile | 95.3 | 97.5 | 96.4 | 10.2 | 93.9 | 98.9 | 96.3 | 10.4 | nan | nan | nan | nan |
| BonusValue | 95.0 | 100 | 97.5 | 88.1 | 97.4 | 100 | 98.7 | 82.1 | nan | nan | nan | nan |
| BonusSign | 66.3 | 100 | 79.7 | 65.6 | 53.1 | 100 | 69.4 | 100 | nan | nan | nan | nan |

## F.13 CENTIPEDE DETAILS

You are an elf and must use your magic wands to fend off spiders, fleas and centipedes. Your goal is to protect mushrooms in an enchanted forest. If you are bitten by a spider, flea or centipede, you will be temporally paralyzed and you will lose a magic wand. The game ends once you have lost all wands. You may receive additional wands after scoring a sufficient number of points.

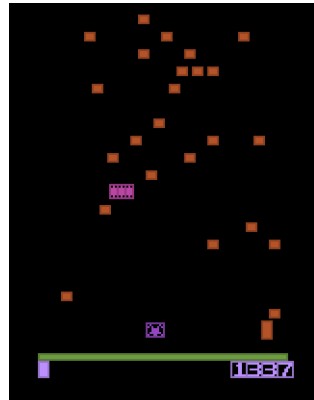

Table 19: Per class statistics on Centipede

| | Random | | | | DQN | | | | C51 | | | |
| | Pr | Rec | F-sc | IOU | Pr | Rec | F-sc | IOU | Pr | Rec | F-sc | IOU |
|---|---|---|---|---|---|---|---|---|---|---|---|---|
| Score | 99.6 | 100 | 99.8 | 97.0 | 98.8 | 100 | 99.4 | 97.7 | nan | nan | nan | nan |
| Projectile | 84.9 | 99.7 | 91.7 | 88.9 | 84.8 | 97.5 | 90.7 | 88.6 | nan | nan | nan | nan |
| Life | 100 | 100 | 100 | 100 | 100 | 100 | 100 | 100 | nan | nan | nan | nan |
| Ground | 100 | 97.8 | 98.9 | 100 | 100 | 98.4 | 99.2 | 100 | nan | nan | nan | nan |
| Mushroom | 99.6 | 99.7 | 99.6 | 99.8 | 99.3 | 99.7 | 99.5 | 99.6 | nan | nan | nan | nan |
| CentipedeSegment | 72.2 | 78.8 | 75.4 | 57.1 | 69.0 | 75.2 | 72.0 | 55.3 | nan | nan | nan | nan |
| Player | 96.4 | 91.8 | 94.1 | 87.4 | 94.4 | 90.8 | 92.6 | 84.8 | nan | nan | nan | nan |
| Spider | 92.4 | 100 | 96.1 | 99.6 | 87.1 | 99.0 | 92.7 | 100 | nan | nan | nan | nan |
| Flea | 50.0 | 100 | 66.7 | 100 | 50.0 | 100 | 66.7 | 100 | nan | nan | nan | nan |
| Scorpion | nan | nan | nan | nan | 100 | 100 | 100 | 100 | nan | nan | nan | nan |

## F.14 CHOPPERCOMMAND DETAILS

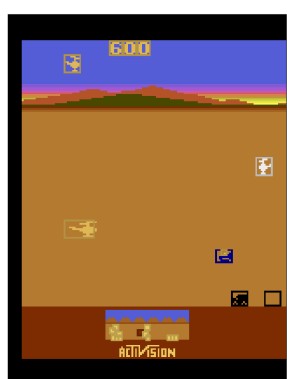

You control a helicopter and must protect truck convoys. To that end, you need to shoot down enemy aircraft.A mini-map is displayed at the bottom of the screen.

Table 20: Per class statistics on ChopperCommand

|  | Random | | | | DQN | | | | C51 | | | |
|---|---|---|---|---|---|---|---|---|---|---|---|---|
|  | Pr | Rec | F-sc | IOU | Pr | Rec | F-sc | IOU | Pr | Rec | F-sc | IOU |
| Score | 100 | 100 | 100 | 100 | 100 | 100 | 100 | 100 | 100 | 100 | 100 | 100 |
| Life | 100 | 100 | 100 | 100 | 100 | 100 | 100 | 100 | 100 | 100 | 100 | 100 |
| MiniEnemy | 88.9 | 78.0 | 83.1 | 48.5 | 66.8 | 62.3 | 64.5 | 51.1 | 47.4 | 39.4 | 43.0 | 56.2 |
| MiniTruck | 86.0 | 93.1 | 89.4 | 100 | 68.5 | 70.6 | 69.6 | 99.9 | 49.4 | 58.5 | 53.6 | 100 |
| Truck | 88.8 | 99.6 | 93.9 | 79.6 | 95.9 | 98.6 | 97.2 | 80.4 | 92.9 | 96.4 | 94.6 | 74.7 |
| MiniPlayer | 90.8 | 100 | 95.2 | 84.8 | 99.2 | 100 | 99.6 | 84.5 | 99.8 | 100 | 99.9 | 93.7 |
| Player | 95.2 | 98.8 | 96.9 | 79.0 | 91.4 | 89.4 | 90.4 | 75.8 | 99.0 | 98.8 | 98.9 | 86.5 |
| Shot | 76.9 | 74.9 | 75.9 | 88.0 | 58.8 | 58.8 | 58.8 | 87.3 | 81.8 | 81.8 | 81.8 | 90.2 |
| EnemyHelicopter | 79.9 | 94.3 | 86.5 | 71.4 | 45.6 | 85.4 | 59.5 | 73.7 | 61.8 | 95.5 | 75.0 | 71.1 |
| Bomb | 51.3 | 93.2 | 66.2 | 37.7 | 54.7 | 80.3 | 65.0 | 55.9 | 50.0 | 82.1 | 62.2 | 72.5 |
| EnemyPlane | 92.0 | 98.3 | 95.0 | 66.6 | 94.0 | 96.2 | 95.1 | 71.4 | 94.4 | 94.4 | 94.4 | 74.4 |

## F.15 CRAZYCLIMBER DETAILS

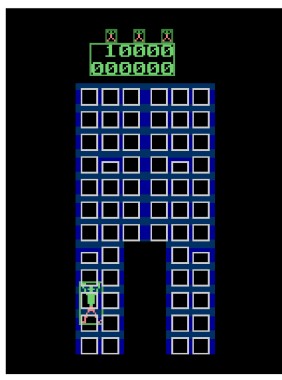

You are a climber trying to reach the top of four buildings, while avoiding obstacles like closing windows and falling objects. When you receive damage (windows closing or objects) you will fall and lose one life; you have a total of 5 lives before the end games. At the top of each building, there's a helicopter which you need to catch to get to the next building. The goal is to climb as fast as possible while receiving the least amount of damage.

Table 21: Per class IOU on CrazyClimber

| | Random | | | | DQN | | | | C51 | | | |
|---|---|---|---|---|---|---|---|---|---|---|---|---|
| | Pr | Rec | F-sc | IOU | Pr | Rec | F-sc | IOU | Pr | Rec | F-sc | IOU |
| Player | 96.0 | 97.6 | 96.8 | 91.1 | 90.8 | 95.0 | 92.8 | 90.4 | nan | nan | nan | nan |
| Window | 97.6 | 95.2 | 96.4 | 97.4 | 98.3 | 94.6 | 96.4 | 96.5 | nan | nan | nan | nan |
| Score | 100 | 100 | 100 | 100 | 100 | 100 | 100 | 100 | nan | nan | nan | nan |
| Life | 100 | 100 | 100 | 100 | 100 | 100 | 100 | 100 | nan | nan | nan | nan |
| Enemy_Red | 75.0 | 81.8 | 78.3 | 55.3 | 19.1 | 78.8 | 30.8 | 57.2 | nan | nan | nan | nan |
| Purple_Projectile | 66.7 | 66.7 | 66.7 | 45.0 | 56.2 | 69.2 | 62.1 | 52.8 | nan | nan | nan | nan |
| Yellow_Projectile | 33.3 | 50.0 | 40.0 | 70.9 | 44.4 | 57.1 | 50.0 | 79.7 | nan | nan | nan | nan |
| Yellow_Ball | 84.0 | 91.3 | 87.5 | 67.8 | 70.5 | 79.5 | 74.7 | 64.8 | nan | nan | nan | nan |
| Enemy_Bird | 69.7 | 92.0 | 79.3 | 76.1 | 58.5 | 79.2 | 67.3 | 65.0 | nan | nan | nan | nan |
| Helicopter | nan | nan | nan | nan | 14.3 | 14.3 | 14.3 | 42.7 | nan | nan | nan | nan |
| Blue_Projectile | nan | nan | nan | nan | 100 | 100 | 100 | 53.6 | nan | nan | nan | nan |

## F.16 DEMONATTACK DETAILS

You are facing waves of demons in the ice planet of Krybor. Points are accumulated by destroying demons. You begin with 3 reserve bunkers, and can increase its number (up to 6) by avoiding enemy attacks. Each attack wave you survive without any hits, grants you a new bunker. Every time an enemy hits you, a bunker is destroyed. When the last bunker falls, the next enemy hit will destroy you and the game ends.

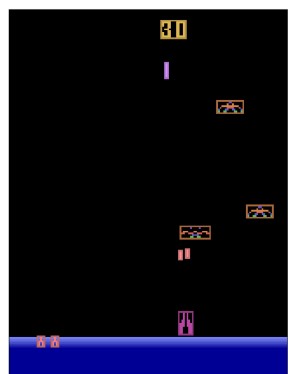

Table 22: Per class IOU on DemonAttack

| | Random | | | | DQN | | | | C51 | | | |
|---|---|---|---|---|---|---|---|---|---|---|---|---|
| | Pr | Rec | F-sc | IOU | Pr | Rec | F-sc | IOU | Pr | Rec | F-sc | IOU |
| ProjectileFriendly | 94.8 | 100 | 97.3 | 83.8 | 97.2 | 99.4 | 98.3 | 85.1 | nan | nan | nan | nan |
| Score | 100 | 98.6 | 99.3 | 97.8 | 98.6 | 91.1 | 94.7 | 95.8 | nan | nan | nan | nan |
| Player | 92.6 | 100 | 96.2 | 100 | 97.2 | 100 | 98.6 | 100 | nan | nan | nan | nan |
| Live | 99.1 | 100 | 99.5 | 100 | 97.5 | 100 | 98.8 | 100 | nan | nan | nan | nan |
| Enemy | 99.3 | 97.9 | 98.6 | 73.7 | 75.1 | 56.6 | 64.5 | 66.3 | nan | nan | nan | nan |
| ProjectileHostile | 80.2 | 98.7 | 88.5 | 55.4 | 59.9 | 98.3 | 74.4 | 37.2 | nan | nan | nan | nan |

## F.17 FISHINGDERBY DETAILS

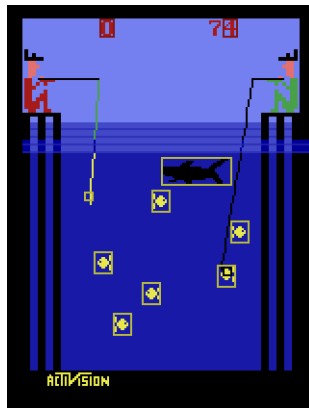

Your objective is to catch more sunfish than your opponent.

Table 23: Per class statistics on FishingDerby

|  | Random | | | | DQN | | | | C51 | | | |
|---|---|---|---|---|---|---|---|---|---|---|---|---|
|  | Pr | Rec | F-sc | IOU | Pr | Rec | F-sc | IOU | Pr | Rec | F-sc | IOU |
| ScorePlayerTwo | 100 | 53.1 | 69.4 | 100 | 100 | 52.5 | 68.9 | 100 | 100 | 52.5 | 68.9 | 100 |
| Fish | 94.5 | 98.2 | 96.3 | 68.7 | 90.6 | 98.5 | 94.4 | 69.2 | 87.1 | 99.3 | 92.8 | 68.8 |
| PlayerTwoHook | 62.6 | 66.2 | 64.3 | 29.4 | 62.8 | 67.7 | 65.1 | 26.5 | 62.0 | 65.1 | 63.5 | 30.9 |
| ScorePlayerOne | 100 | 96.3 | 98.1 | 100 | 100 | 73.5 | 84.7 | 100 | 100 | 52.5 | 68.8 | 100 |
| PlayerOneHook | 69.0 | 69.0 | 69.0 | 22.6 | 87.4 | 87.6 | 87.5 | 22.5 | 53.8 | 56.9 | 55.3 | 22.4 |
| Shark | 82.4 | 99.8 | 90.2 | 92.1 | 82.8 | 99.5 | 90.4 | 91.5 | 77.0 | 99.7 | 86.9 | 92.7 |

## F.18 FREEWAY DETAILS

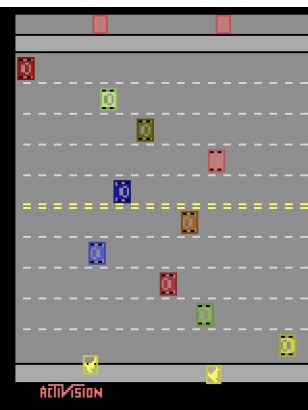

Your objective is to guide your chicken across lane after lane of busy rush hour traffic. You receive a point for every chicken that makes it to the top of the screen after crossing all the lanes of traffic.

Table 24: Per class statistics on Freeway

|  | Random | | | | DQN | | | | C51 | | | |
|---|---|---|---|---|---|---|---|---|---|---|---|---|
|  | Pr | Rec | F-sc | IOU | Pr | Rec | F-sc | IOU | Pr | Rec | F-sc | IOU |
| Chicken | 100 | 100 | 100 | 97.1 | 99.8 | 99.9 | 99.8 | 96.9 | 97.0 | 98.5 | 97.7 | 96.4 |
| Car | 99.1 | 99.9 | 99.5 | 87.0 | 99.3 | 99.9 | 99.6 | 87.0 | 99.4 | 99.8 | 99.6 | 87.0 |
| Score | 95.0 | 48.8 | 64.5 | 100 | 93.4 | 48.4 | 63.7 | 100 | 85.0 | 52.3 | 64.8 | 86.1 |

### F.19   FROSTBITE DETAILS

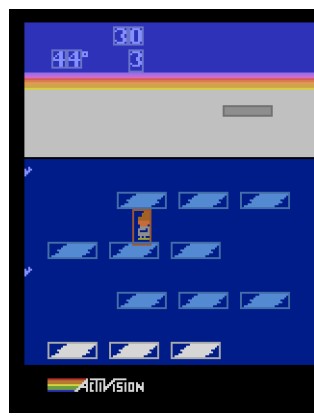

In Frostbite, the player controls "Frostbite Bailey" who hops back and forth across an Arctic river, changing the color of the ice blocks from white to blue. Each time he does so, a block is added to his igloo.

Table 25: Per class statistics on Frostbite

|  | Random | | | | DQN | | | | C51 | | | |
|---|---|---|---|---|---|---|---|---|---|---|---|---|
|  | Pr | Rec | F-sc | IOU | Pr | Rec | F-sc | IOU | Pr | Rec | F-sc | IOU |
| WhitePlate | 99.5 | 99.7 | 99.6 | 92.0 | 93.2 | 99.8 | 96.4 | 88.5 | 92.0 | 98.8 | 95.3 | 87.1 |
| Degree | 100 | 100 | 100 | 98.0 | 100 | 100 | 100 | 97.2 | 100 | 100 | 100 | 97.7 |
| PlayerScore | 100 | 100 | 100 | 96.0 | 100 | 100 | 100 | 89.7 | 100 | 100 | 100 | 89.8 |
| Player | 63.8 | 100 | 77.9 | 71.4 | 66.8 | 100 | 80.1 | 72.1 | 75.8 | 100 | 86.2 | 70.9 |
| LifeCount | 100 | 100 | 100 | 89.0 | 100 | 100 | 100 | 87.2 | 100 | 100 | 100 | 87.9 |
| House | 100 | 100 | 100 | 99.6 | 99.7 | 100 | 99.9 | 97.1 | 100 | 100 | 100 | 94.6 |
| BluePlate | 98.8 | 99.5 | 99.1 | 91.6 | 78.0 | 100 | 87.6 | 84.7 | 74.0 | 98.2 | 84.4 | 82.8 |
| Bird | 98.2 | 95.3 | 96.7 | 99.1 | 90.1 | 79.7 | 84.6 | 95.6 | 91.4 | 84.4 | 87.7 | 94.7 |
| CompletedHouse | nan | nan | nan | nan | 100 | 100 | 100 | 99.6 | 100 | 100 | 100 | 99.8 |
| GreenFish | nan | nan | nan | nan | 82.2 | 77.8 | 79.9 | 67.2 | 84.2 | 87.8 | 86.0 | 70.1 |
| Crab | nan | nan | nan | nan | 96.4 | 78.4 | 86.5 | 66.3 | 96.6 | 81.1 | 88.2 | 63.5 |
| Bear | nan | nan | nan | nan | 90.0 | 90.0 | 90.0 | 77.3 | 100 | 91.4 | 95.5 | 80.6 |
| Clam | nan | nan | nan | nan | 59.5 | 97.8 | 73.9 | 75.1 | 66.7 | 100 | 80.0 | 58.3 |

### F.20   GOPHER DETAILS

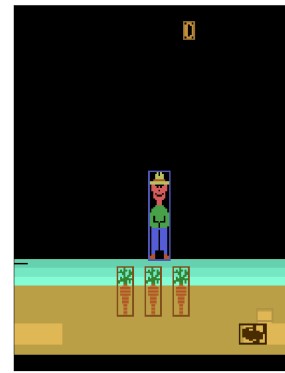

The player controls a shovel-wielding farmer who protects a crop of three carrots from a gopher.

Table 26: Per class IOU on Gopher

|  | Random | | | | DQN | | | | C51 | | | |
|  | Pr | Rec | F-sc | IOU | Pr | Rec | F-sc | IOU | Pr | Rec | F-sc | IOU |
|---|---|---|---|---|---|---|---|---|---|---|---|---|
| Gopher | 74.8 | 97.7 | 84.7 | 84.7 | 71.8 | 96.0 | 82.2 | 83.3 | nan | nan | nan | nan |
| Score | 100 | 100 | 100 | 98.4 | 100 | 96.9 | 98.4 | 93.1 | nan | nan | nan | nan |
| Player | 99.8 | 99.8 | 99.8 | 81.7 | 100 | 100 | 100 | 80.3 | nan | nan | nan | nan |
| Empty_Block | 100 | 35.0 | 51.8 | 65.6 | 100 | 24.6 | 39.5 | 62.5 | nan | nan | nan | nan |
| Carrot | 100 | 100 | 100 | 100 | 99.9 | 100 | 100 | 100 | nan | nan | nan | nan |
| Bird | nan | nan | nan | nan | 33.8 | 98.0 | 50.3 | 84.8 | nan | nan | nan | nan |

## F.21  HERO DETAILS

You need to rescue miners that are stuck in a mine shaft. You have access to various tools: A propeller backpack that allows you to fly wherever you want, sticks of dynamite that can be used to blast through walls, a laser beam to kill vermin, and a raft to float across stretches of lava. You have a limited amount of power. Once you run out, you lose a live.

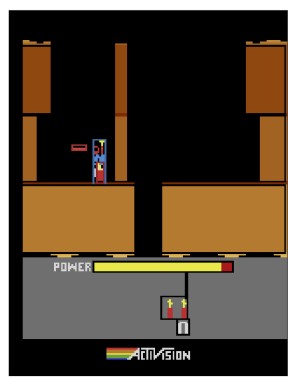

Table 27: Per class IOU on Hero

|  | Random | | | | DQN | | | | C51 | | | |
|  | Pr | Rec | F-sc | IOU | Pr | Rec | F-sc | IOU | Pr | Rec | F-sc | IOU |
|---|---|---|---|---|---|---|---|---|---|---|---|---|
| Life | 65.0 | 100 | 78.8 | 71.1 | 84.8 | 100 | 91.8 | 75.6 | 81.4 | 100 | 89.7 | 78.7 |
| Score | 65.8 | 70.6 | 68.1 | 48.7 | 54.4 | 62.0 | 57.9 | 57.6 | 76.8 | 82.9 | 79.8 | 59.3 |
| Player | 96.0 | 99.8 | 97.9 | 96.7 | 91.0 | 99.8 | 95.2 | 99.4 | 94.0 | 100 | 96.9 | 99.5 |
| PowerBar | 100 | 100 | 100 | 99.8 | 100 | 100 | 100 | 99.8 | 100 | 100 | 100 | 99.9 |
| BombStock | 81.8 | 100 | 90.0 | 81.0 | 98.8 | 100 | 99.4 | 90.1 | 99.4 | 100 | 99.7 | 80.9 |
| Wall | 100 | 93.6 | 96.7 | 98.5 | 73.2 | 91.4 | 81.3 | 91.8 | 83.9 | 96.4 | 89.7 | 95.0 |
| LaserBeam | 36.8 | 84.2 | 51.2 | 35.9 | 19.8 | 81.4 | 31.8 | 34.2 | 15.0 | 81.8 | 25.3 | 33.0 |
| Bomb | 42.3 | 81.1 | 55.6 | 83.3 | 45.9 | 28.3 | 35.0 | 83.3 | 41.3 | 18.7 | 25.7 | 75.3 |
| Enemy | 94.1 | 100 | 97.0 | 33.2 | 37.0 | 71.8 | 48.8 | 43.0 | 39.9 | 64.9 | 49.4 | 39.0 |
| EndNPC | 100 | 100 | 100 | 69.2 | 100 | 83.8 | 91.2 | 69.2 | 100 | 77.5 | 87.3 | 69.2 |
| Lamp | nan | nan | nan | nan | 46.6 | 100 | 63.6 | 62.5 | 82.0 | 100 | 90.1 | 62.5 |
| LavaWall | nan | nan | nan | nan | 45.3 | 77.4 | 57.1 | 92.3 | 54.4 | 82.8 | 65.7 | 99.2 |

## F.22    IceHockey details

Your goal is to score as many points as possible in a standard game of Ice Hockey over a 3-minute time period. The ball is usually called "the puck".There are 32 shot angles ranging from the extreme left to the extreme right. The angles can only aim towards the opponent's goal.Just as in real hockey, you can pass the puck by shooting it off the sides of the rink. This can be really key when you're in position to score a goal.

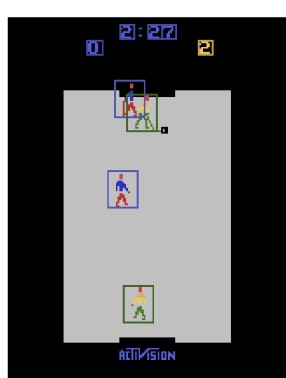

Table 28: Per class IOU on IceHockey

|  | Random | | | | DQN | | | | C51 | | | |
|---|---|---|---|---|---|---|---|---|---|---|---|---|
|  | Pr | Rec | F-sc | IOU | Pr | Rec | F-sc | IOU | Pr | Rec | F-sc | IOU |
| EnemyScore | 71.6 | 100 | 83.4 | 76.4 | 57.1 | 100 | 72.7 | 76.4 | nan | nan | nan | nan |
| Player | 99.1 | 99.1 | 99.1 | 47.8 | 98.2 | 98.2 | 98.2 | 47.9 | nan | nan | nan | nan |
| Enemy | 99.0 | 99.5 | 99.2 | 52.4 | 99.6 | 99.6 | 99.6 | 52.8 | nan | nan | nan | nan |
| PlayerScore | 83.8 | 100 | 91.2 | 67.7 | 65.3 | 100 | 79.0 | 67.7 | nan | nan | nan | nan |
| Ball | 85.2 | 98.6 | 91.4 | 80.2 | 83.4 | 98.8 | 90.5 | 84.1 | nan | nan | nan | nan |
| Timer | 100 | 100 | 100 | 80.5 | 100 | 100 | 100 | 79.7 | nan | nan | nan | nan |

## F.23    Kangaroo details

The object of the game is to score as many points as you can while controlling Mother Kangaroo to rescue her precious baby. You start the game with three lives.During this rescue mission, Mother Kangaroo encounters many obstacles. You need to help her climb ladders, pick bonus fruit, and throw punches at monkeys.

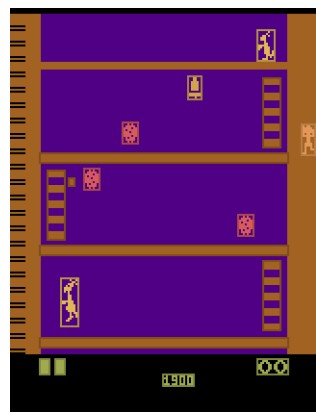

Table 29: Per class statistics on Kangaroo

| | Random | | | | DQN | | | | C51 | | | |
|---|---|---|---|---|---|---|---|---|---|---|---|---|
| | Pr | Rec | F-sc | IOU | Pr | Rec | F-sc | IOU | Pr | Rec | F-sc | IOU |
| Bell | 99.8 | 99.8 | 99.8 | 100 | 100 | 100 | 100 | 100 | 99.8 | 99.8 | 99.8 | 100 |
| Platform | 100 | 75.0 | 85.7 | 100 | 100 | 75.0 | 85.7 | 100 | 100 | 75.0 | 85.7 | 100 |
| Scale | 100 | 100 | 100 | 100 | 100 | 100 | 100 | 100 | 100 | 100 | 100 | 100 |
| Fruit | 99.8 | 99.9 | 99.9 | 90.7 | 99.2 | 100 | 99.6 | 90.9 | 97.4 | 99.8 | 98.6 | 90.7 |
| Child | 99.6 | 99.8 | 99.7 | 95.0 | 100 | 100 | 100 | 95.6 | 99.8 | 100 | 99.9 | 96.4 |
| Life | 99.7 | 100 | 99.8 | 100 | 100 | 100 | 100 | 100 | 99.6 | 100 | 99.8 | 100 |
| Score | 99.8 | 100 | 99.9 | 100 | 100 | 100 | 100 | 99.6 | 99.8 | 100 | 99.9 | 99.3 |
| Time | 99.8 | 100 | 99.9 | 100 | 100 | 100 | 100 | 89.7 | 99.8 | 100 | 99.9 | 96.5 |
| Player | 79.8 | 89.3 | 84.3 | 79.4 | 88.0 | 91.7 | 89.8 | 78.6 | 81.2 | 88.3 | 84.6 | 79.3 |
| Projectile_top | 81.6 | 84.7 | 83.1 | 81.0 | 98.9 | 99.5 | 99.2 | 87.6 | 92.9 | 88.3 | 90.5 | 85.5 |
| Enemy | 87.2 | 95.5 | 91.2 | 91.1 | 94.5 | 95.2 | 94.8 | 86.8 | 85.8 | 96.4 | 90.8 | 86.8 |
| Projectile_enemy | 66.7 | 93.3 | 77.8 | 33.3 | 50.0 | 100 | 66.7 | 33.3 | 14.1 | 86.7 | 24.2 | 33.3 |

## F.24 MONTEZUMAREVENGE DETAILS

Your goal is to acquire Montezuma's treasure by making your way through a maze of chambers within the emperor's fortress. You must avoid deadly creatures while collecting valuables and tools which can help you escape with the treasure.

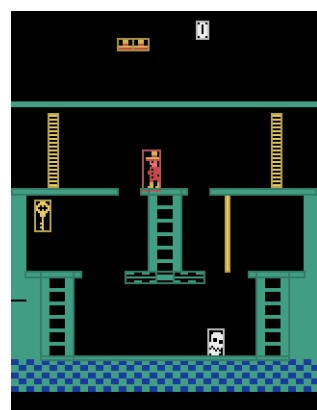

Table 30: Per class statistics on MontezumaRevenge

| | Random | | | | DQN | | | | C51 | | | |
|---|---|---|---|---|---|---|---|---|---|---|---|---|
| | Pr | Rec | F-sc | IOU | Pr | Rec | F-sc | IOU | Pr | Rec | F-sc | IOU |
| Skull | 99.6 | 99.6 | 99.6 | 79.0 | 100 | 100 | 100 | 79.1 | 100 | 100 | 100 | 80.4 |
| Life | 100 | 100 | 100 | 100 | 100 | 100 | 100 | 100 | 100 | 100 | 100 | 100 |
| Player | 99.0 | 98.6 | 98.8 | 77.9 | 100 | 100 | 100 | 97.3 | 100 | 100 | 100 | 100 |
| Rope | 97.6 | 100 | 98.8 | 100 | 100 | 100 | 100 | 100 | 100 | 100 | 100 | 100 |
| Barrier | 100 | 100 | 100 | 100 | 100 | 100 | 100 | 100 | 100 | 100 | 100 | 100 |
| Key | 99.0 | 96.7 | 97.8 | 100 | 100 | 100 | 100 | 100 | 100 | 100 | 100 | 100 |
| Score | 100 | 100 | 100 | 100 | 100 | 100 | 100 | 100 | 100 | 100 | 100 | 100 |

### F.25    MsPacman details

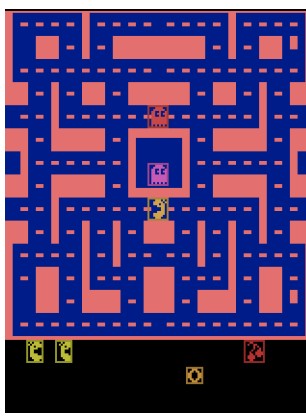

Your goal is to collect all of the pellets on the screen while avoiding the ghosts.

Table 31: Per class statistics on MsPacman

|        | Random |      |      |      | DQN  |      |      |      | C51 |     |      |     |
|--------|--------|------|------|------|------|------|------|------|-----|-----|------|-----|
|        | Pr     | Rec  | F-sc | IOU  | Pr   | Rec  | F-sc | IOU  | Pr  | Rec | F-sc | IOU |
| Life   | 100    | 100  | 100  | 93.3 | 54.4 | 100  | 70.5 | 90.3 | nan | nan | nan  | nan |
| Score  | 100    | 100  | 100  | 98.2 | 100  | 100  | 100  | 99.2 | nan | nan | nan  | nan |
| Player | 99.8   | 99.8 | 99.8 | 71.7 | 94.8 | 99.0 | 96.8 | 72.5 | nan | nan | nan  | nan |
| Ghost  | 55.6   | 98.4 | 71.1 | 79.5 | 61.3 | 98.3 | 75.5 | 77.2 | nan | nan | nan  | nan |
| Fruit  | 100    | 100  | 100  | 88.9 | 97.9 | 100  | 98.9 | 85.5 | nan | nan | nan  | nan |

### F.26    Pitfall details

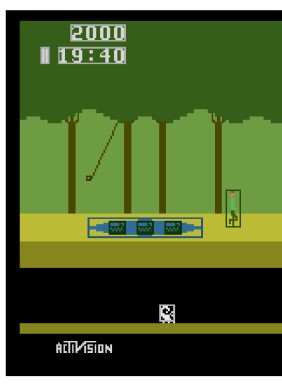

You control Pitfall Harry and are tasked with collecting all the treasures in a jungle within 20 minutes. You have three lives. The game is over if you collect all the treasures or if you die or if the time runs out.

Table 32: Per class IOU on Pitfall

| | Random | | | | DQN | | | | C51 | | | |
|---|---|---|---|---|---|---|---|---|---|---|---|---|
| | Pr | Rec | F-sc | IOU | Pr | Rec | F-sc | IOU | Pr | Rec | F-sc | IOU |
| Player | 91.8 | 97.0 | 94.3 | 68.7 | 100 | 100 | 100 | 61.2 | nan | nan | nan | nan |
| Wall | 100 | 100 | 100 | 99.8 | 100 | 100 | 100 | 76.5 | nan | nan | nan | nan |
| Logs | 99.1 | 99.8 | 99.5 | 99.6 | 100 | 100 | 100 | 98.9 | nan | nan | nan | nan |
| LifeCount | 100 | 100 | 100 | 100 | 100 | 100 | 100 | 100 | nan | nan | nan | nan |
| Timer | 100 | 100 | 100 | 99.7 | 100 | 100 | 100 | 98.2 | nan | nan | nan | nan |
| StairPit | 100 | 100 | 100 | 100 | 100 | 100 | 100 | 100 | nan | nan | nan | nan |
| PlayerScore | 100 | 100 | 100 | 97.0 | 100 | 100 | 100 | 96.7 | nan | nan | nan | nan |
| Scorpion | 100 | 100 | 100 | 94.7 | 100 | 100 | 100 | 100 | nan | nan | nan | nan |
| Waterhole | 96.7 | 100 | 98.3 | 100 | 73.3 | 100 | 84.6 | 99.6 | nan | nan | nan | nan |
| Crocodile | 100 | 100 | 100 | 100 | nan | nan | nan | nan | nan | nan | nan | nan |
| Rope | 87.7 | 69.4 | 77.5 | 100 | nan | nan | nan | nan | nan | nan | nan | nan |
| Snake | 100 | 100 | 100 | 97.6 | 100 | 100 | 100 | 96.9 | nan | nan | nan | nan |
| Tarpit | 80.0 | 100 | 88.9 | 100 | nan | nan | nan | nan | nan | nan | nan | nan |

## F.27 PONG DETAILS

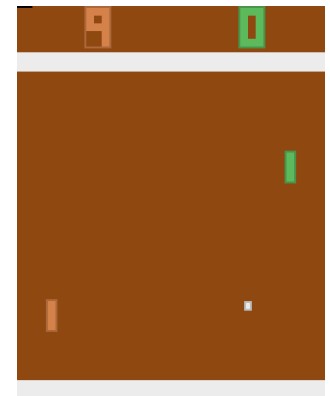

You control the right paddle, you compete against the left paddle controlled by the computer. You each try to keep deflecting the ball away from your goal and into your opponent's goal.

Table 33: Per class statistics on Pong

| | Random | | | | DQN | | | | C51 | | | |
|---|---|---|---|---|---|---|---|---|---|---|---|---|
| | Pr | Rec | F-sc | IOU | Pr | Rec | F-sc | IOU | Pr | Rec | F-sc | IOU |
| Ball | 60.2 | 100 | 75.2 | 74.8 | 76.0 | 100 | 86.4 | 75.2 | 74.0 | 100 | 85.1 | 74.9 |
| Player | 100 | 100 | 100 | 91.7 | 100 | 100 | 100 | 89.0 | 100 | 100 | 100 | 94.7 |
| EnemyScore | 100 | 96.9 | 98.4 | 78.0 | 100 | 95.9 | 97.9 | 79.0 | 100 | 94.7 | 97.3 | 79.2 |
| Enemy | 85.2 | 100 | 92.0 | 94.4 | 92.8 | 100 | 96.3 | 94.2 | 92.2 | 100 | 95.9 | 94.1 |
| PlayerScore | 100 | 100 | 100 | 74.3 | 100 | 100 | 100 | 84.1 | 100 | 95.1 | 97.5 | 86.5 |

## F.28 PRIVATEEYE DETAILS

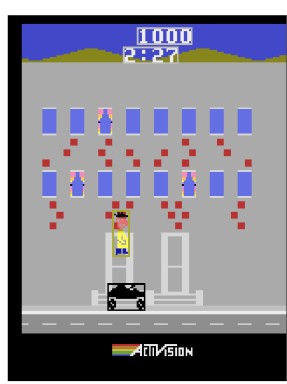

You control the French Private Eye Pierre Touche. Navigate the city streets, parks, secret passages, dead-ends and one-ways in search of the ringleader, Henri Le Fiend and his gang. You also need to find evidence and stolen goods that are scattered about. There are five cases, complete each case before its statute of limitations expires.

Table 34: Per class IOU on PrivateEye

|        | Random |     |      |      | DQN  |     |      |      | C51 |     |      |     |
|        | Pr   | Rec | F-sc | IOU  | Pr   | Rec  | F-sc | IOU  | Pr  | Rec | F-sc | IOU |
|--------|------|-----|------|------|------|------|------|------|-----|-----|------|-----|
| Car    | 96.8 | 100 | 98.4 | 97.3 | 100  | 100  | 100  | 94.4 | nan | nan | nan  | nan |
| Clock  | 100  | 100 | 100  | 98.5 | 100  | 100  | 100  | 98.6 | nan | nan | nan  | nan |
| Player | 95.8 | 97.0| 96.4 | 95.6 | 99.2 | 99.2 | 99.2 | 92.8 | nan | nan | nan  | nan |
| Score  | 100  | 100 | 100  | 97.5 | 100  | 100  | 100  | 96.7 | nan | nan | nan  | nan |
| Clue   | 57.5 | 100 | 73.0 | 77.6 | 50.0 | 100  | 66.7 | 78.6 | nan | nan | nan  | nan |
| Mud    | 91.2 | 100 | 95.4 | 100  | nan  | nan  | nan  | nan  | nan | nan | nan  | nan |
| Dove   | 100  | 100 | 100  | 100  | nan  | nan  | nan  | nan  | nan | nan | nan  | nan |

## F.29 QBERT DETAILS

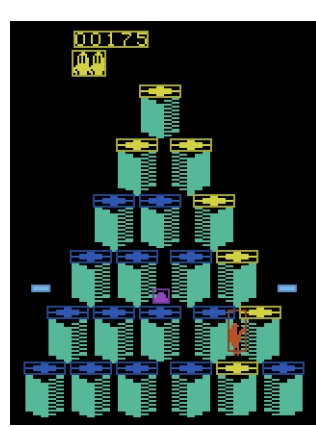

You are Q*bert. Your goal is to change the color of all the cubes on the pyramid to the pyramid's 'destination' color. To do this, you must hop on each cube on the pyramid one at a time while avoiding nasty creatures that lurk there.

Table 35: Per class statistics on Qbert

| | Random | | | | DQN | | | | C51 | | | |
| | Pr | Rec | F-sc | IOU | Pr | Rec | F-sc | IOU | Pr | Rec | F-sc | IOU |
|---|---|---|---|---|---|---|---|---|---|---|---|---|
| Cube | 100 | 99.6 | 99.8 | 100 | 74.6 | 99.5 | 85.3 | 99.9 | 77.6 | 99.5 | 87.2 | 99.9 |
| Score | 66.0 | 100 | 79.5 | 100 | 34.8 | 100 | 51.6 | 98.8 | 31.8 | 100 | 48.3 | 98.9 |
| Lives | 60.1 | 100 | 75.0 | 100 | 38.9 | 100 | 56.0 | 100 | 40.5 | 100 | 57.7 | 100 |
| Disk | 99.8 | 100 | 99.9 | 100 | 99.6 | 100 | 99.8 | 100 | 100 | 100 | 100 | 100 |
| Player | 47.3 | 81.9 | 60.0 | 79.2 | 93.7 | 82.1 | 87.5 | 78.2 | 98.3 | 84.1 | 90.6 | 79.9 |
| Sam | 34.5 | 82.9 | 48.7 | 93.8 | 100 | 80.0 | 88.9 | 94.7 | 100 | 93.8 | 96.8 | 95.1 |
| PurpleBall | 20.9 | 52.7 | 29.9 | 90.5 | 71.3 | 69.0 | 70.2 | 91.3 | 76.1 | 66.4 | 70.9 | 90.9 |
| Coily | 91.3 | 100 | 95.5 | 93.4 | 91.3 | 100 | 95.5 | 89.4 | 87.1 | 100 | 93.1 | 90.7 |
| GreenBall | nan | nan | nan | nan | 83.3 | 90.9 | 87.0 | 87.7 | 79.3 | 88.5 | 83.6 | 87.1 |

## F.30 RIVERRAID DETAILS

You control a jet that flies over a river: you can move it sideways and fire missiles to destroy enemy objects. Each time an enemy object is destroyed you score points (i.e. rewards). You lose a jet when you run out of fuel: fly over a fuel depot when you begin to run low. You lose a jet even when it collides with the river bank or one of the enemy objects (except fuel depots). The game begins with a squadron of three jets in reserve and you're given an additional jet (up to 9) for each 10,000 points you score.

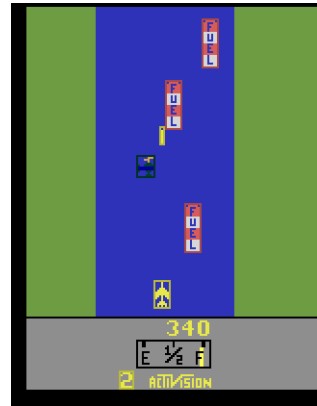

Table 36: Per class statistics on Riverraid

| | Random | | | | DQN | | | | C51 | | | |
| | Pr | Rec | F-sc | IOU | Pr | Rec | F-sc | IOU | Pr | Rec | F-sc | IOU |
|---|---|---|---|---|---|---|---|---|---|---|---|---|
| PlayerScore | 100 | 2.4 | 4.7 | 100 | 100 | 1.0 | 2.0 | 100 | nan | nan | nan | nan |
| FuelDepot | 97.7 | 98.3 | 98.0 | 100 | 94.7 | 96.5 | 95.6 | 100 | nan | nan | nan | nan |
| Tanker | 97.0 | 98.2 | 97.6 | 96.9 | 94.1 | 95.8 | 95.0 | 94.3 | nan | nan | nan | nan |
| Lives | 89.3 | 99.8 | 94.3 | 89.8 | 59.6 | 100 | 74.7 | 92.0 | nan | nan | nan | nan |
| Player | 100 | 99.3 | 99.6 | 76.7 | 99.8 | 99.6 | 99.7 | 74.8 | nan | nan | nan | nan |
| Helicopter | 96.4 | 97.0 | 96.7 | 97.6 | 97.4 | 96.8 | 97.1 | 96.5 | nan | nan | nan | nan |
| PlayerMissile | 87.3 | 92.8 | 90.0 | 88.1 | 82.8 | 95.5 | 88.7 | 89.6 | nan | nan | nan | nan |
| Bridge | 95.2 | 100 | 97.6 | 82.7 | 97.6 | 97.6 | 97.6 | 84.9 | nan | nan | nan | nan |
| Jet | nan | nan | nan | nan | 95.9 | 100 | 97.9 | 83.6 | nan | nan | nan | nan |

### F.31 ROADRUNNER DETAILS

You control the Road Runner(TM) in a race; you can control the direction to run in and times to jumps.The goal is to outrun Wile E. Coyote(TM) while avoiding the hazards of the desert.The game begins with three lives. You lose a life when the coyote catches you, picks you up in a rocket, or shoots you with a cannon. You also lose a life when a truck hits you, you hit a land mine, you fall off a cliff,or you get hit by a falling rock.You score points (i.e. rewards) by eating seeds along the road, eating steel shot, and destroying the coyote.

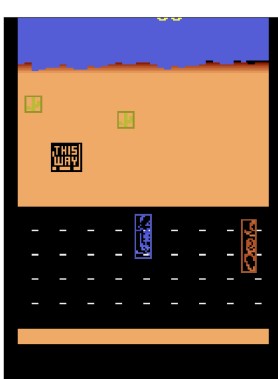

Table 37: Per class statistics on RoadRunner

|  | Random | | | | DQN | | | | C51 | | | |
|  | Pr | Rec | F-sc | IOU | Pr | Rec | F-sc | IOU | Pr | Rec | F-sc | IOU |
|---|---|---|---|---|---|---|---|---|---|---|---|---|
| Enemy | 99.3 | 88.6 | 93.7 | 77.1 | 83.3 | 85.3 | 84.3 | 71.7 | nan | nan | nan | nan |
| Sign | 94.4 | 100 | 97.1 | 97.8 | 54.3 | 100 | 70.4 | 90.9 | nan | nan | nan | nan |
| Cactus | 99.5 | 99.0 | 99.2 | 98.4 | 91.3 | 93.8 | 92.6 | 89.5 | nan | nan | nan | nan |
| Bird | 100 | 100 | 100 | 100 | 97.4 | 97.2 | 97.3 | 100 | nan | nan | nan | nan |
| Player | 95.4 | 96.8 | 96.1 | 79.4 | 93.4 | 98.7 | 96.0 | 91.4 | nan | nan | nan | nan |
| BirdSeeds | 55.2 | 90.6 | 68.6 | 78.3 | 62.3 | 89.1 | 73.3 | 63.0 | nan | nan | nan | nan |
| Truck | nan | nan | nan | nan | 77.4 | 100 | 87.3 | 72.0 | nan | nan | nan | nan |
| RoadCrack | nan | nan | nan | nan | 87.5 | 8.2 | 15.1 | 72.5 | nan | nan | nan | nan |
| AcmeMine | nan | nan | nan | nan | 16.7 | 41.7 | 23.8 | 69.0 | nan | nan | nan | nan |

### F.32 SEAQUEST DETAILS

You control a sub able to move in all directions and fire torpedoes. The goal is to retrieve as many divers as you can, while dodging and blasting enemy subs and killer sharks; points will be awarded accordingly. The game begins with one sub and three waiting on the horizon. Each time you increase your score by 10,000 points, an extra sub will be delivered to your base. You can only have six reserve subs on the screen at one time.Your sub will explode if it collides with anything except your own divers.The sub has a limited amount of oxygen that decreases at a constant rate during the game. When the oxygen tank is almost empty, you need to surface and if you don't do it in time, your sub will blow up and you'll lose one diver. Each time you're forced to surface, with less than six divers, you lose one diver as well.

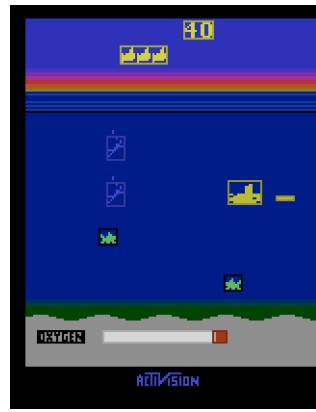

Table 38: Per class statistics on Seaquest

| | Random | | | | DQN | | | | C51 | | | |
| --- | --- | --- | --- | --- | --- | --- | --- | --- | --- | --- | --- | --- |
| | Pr | Rec | F-sc | IOU | Pr | Rec | F-sc | IOU | Pr | Rec | F-sc | IOU |
| OxygenBarDepleted | 92.1 | 100 | 95.9 | 99.8 | 98.6 | 100 | 99.3 | 100 | 98.4 | 100 | 99.2 | 99.9 |
| Player | 75.8 | 98.7 | 85.7 | 75.0 | 95.8 | 98.6 | 97.2 | 80.4 | 95.4 | 99.2 | 97.2 | 80.6 |
| Logo | 100 | 100 | 100 | 100 | 100 | 100 | 100 | 100 | 100 | 100 | 100 | 100 |
| Lives | 100 | 100 | 100 | 100 | 100 | 100 | 100 | 100 | 100 | 100 | 100 | 100 |
| OxygenBarLogo | 99.0 | 97.2 | 98.1 | 100 | 91.4 | 84.0 | 87.5 | 100 | 92.4 | 85.4 | 88.8 | 100 |
| PlayerScore | 100 | 84.0 | 91.3 | 92.9 | 94.0 | 79.4 | 86.1 | 93.6 | 84.8 | 60.1 | 70.4 | 79.9 |
| OxygenBar | 98.9 | 100 | 99.4 | 100 | 91.2 | 100 | 95.4 | 100 | 92.1 | 100 | 95.9 | 100 |
| Diver | 90.6 | 98.2 | 94.3 | 76.9 | 93.8 | 100 | 96.8 | 77.4 | 94.1 | 99.4 | 96.7 | 79.0 |
| PlayerMissile | 70.0 | 100 | 82.4 | 100 | 67.8 | 100 | 80.8 | 100 | 69.1 | 100 | 81.7 | 100 |
| Enemy | 97.8 | 98.9 | 98.3 | 55.8 | 57.8 | 90.6 | 70.6 | 51.5 | 75.6 | 87.5 | 81.1 | 52.7 |
| EnemyMissile | 94.3 | 44.0 | 60.0 | 71.7 | 88.9 | 85.1 | 87.0 | 70.9 | 93.2 | 71.4 | 80.9 | 70.9 |
| CollectedDiver | 100 | 100 | 100 | 100 | 100 | 100 | 100 | 100 | 91.8 | 100 | 95.7 | 100 |
| EnemySubmarine | 95.9 | 100 | 97.9 | 77.6 | 98.1 | 100 | 99.0 | 75.1 | 94.3 | 96.1 | 95.2 | 70.7 |

## F.33 SKIING DETAILS

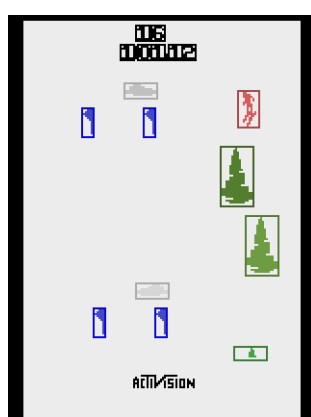

You control a skier who can move sideways. The goal is to run through all gates (between the poles) in the fastest time. You are penalized five seconds for each gate you miss. If you hit a gate or a tree, your skier will jump back up and keep going.

Table 39: Per class statistics on Skiing

| | Random | | | | DQN | | | | C51 | | | |
| --- | --- | --- | --- | --- | --- | --- | --- | --- | --- | --- | --- | --- |
| | Pr | Rec | F-sc | IOU | Pr | Rec | F-sc | IOU | Pr | Rec | F-sc | IOU |
| Score | 100 | 100 | 100 | 100 | 100 | 100 | 100 | 100 | nan | nan | nan | nan |
| Mogul | 97.2 | 99.2 | 98.2 | 84.0 | 100 | 100 | 100 | 85.7 | nan | nan | nan | nan |
| Tree | 81.0 | 83.5 | 82.2 | 75.0 | 51.1 | 51.3 | 51.2 | 48.3 | nan | nan | nan | nan |
| Logo | 100 | 100 | 100 | 100 | 100 | 100 | 100 | 100 | nan | nan | nan | nan |
| Clock | 100 | 100 | 100 | 100 | 100 | 100 | 100 | 100 | nan | nan | nan | nan |
| Player | 99.2 | 96.7 | 97.9 | 64.0 | 100 | 100 | 100 | 63.9 | nan | nan | nan | nan |
| Flag | 95.8 | 97.4 | 96.6 | 86.0 | 100 | 100 | 100 | 86.7 | nan | nan | nan | nan |

### F.34    SPACEINVADERS DETAILS

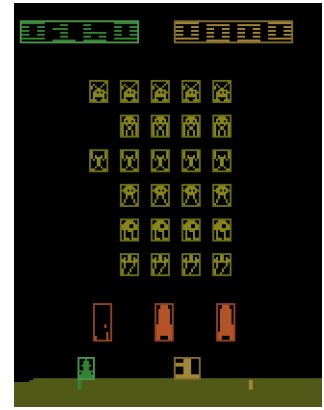

Your objective is to destroy the space invaders by shooting your laser cannon at them before they reach the Earth. The game ends when all your lives are lost after taking enemy fire, or when they reach the earth.

Table 40: Per class statistics on SpaceInvaders

|  | Random | | | | DQN | | | | C51 | | | |
|---|---|---|---|---|---|---|---|---|---|---|---|---|
|  | Pr | Rec | F-sc | IOU | Pr | Rec | F-sc | IOU | Pr | Rec | F-sc | IOU |
| Shield | 98.9 | 98.8 | 98.8 | 90.7 | 100 | 88.3 | 93.8 | 97.6 | nan | nan | nan | nan |
| Score | 79.0 | 100 | 88.3 | 100 | 65.3 | 100 | 79.0 | 100 | nan | nan | nan | nan |
| Lives | 76.8 | 79.1 | 77.9 | 100 | 73.3 | 70.2 | 71.7 | 100 | nan | nan | nan | nan |
| Player | 93.4 | 100 | 96.6 | 91.6 | 94.4 | 100 | 97.1 | 91.9 | nan | nan | nan | nan |
| Alien | 100 | 99.6 | 99.8 | 98.3 | 100 | 98.4 | 99.2 | 99.1 | nan | nan | nan | nan |
| Bullet | 33.5 | 66.1 | 44.5 | 79.2 | 31.9 | 64.1 | 42.6 | 76.8 | nan | nan | nan | nan |
| Satellite | 97.3 | 100 | 98.6 | 93.4 | 100 | 100 | 100 | 92.2 | nan | nan | nan | nan |

### F.35    TENNIS DETAILS

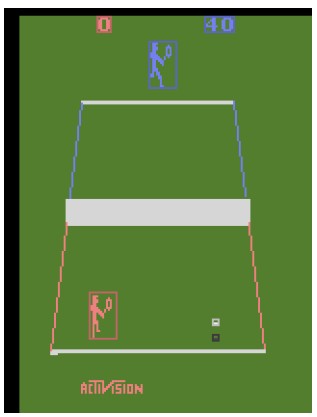

You control the orange player playing against a computer-controlled blue player. The game follows the rules of tennis. The first player to win at least 6 games with a margin of at least two games wins the match. If the score is tied at 6-6, the first player to go 2 games up wins the match.

Table 41: Per class statistics on Tennis

|  | Random | | | | DQN | | | | C51 | | | |
|---|---|---|---|---|---|---|---|---|---|---|---|---|
|  | Pr | Rec | F-sc | IOU | Pr | Rec | F-sc | IOU | Pr | Rec | F-sc | IOU |
| Logo | 100 | 100 | 100 | 100 | 100 | 100 | 100 | 100 | nan | nan | nan | nan |
| EnemyScore | 100 | 100 | 100 | 98.7 | 100 | 100 | 100 | 100 | nan | nan | nan | nan |
| BallShadow | 95.5 | 98.3 | 96.9 | 65.3 | 96.1 | 97.1 | 96.6 | 50.5 | nan | nan | nan | nan |
| Ball | 95.2 | 100 | 97.6 | 70.9 | 95.0 | 100 | 97.4 | 70.3 | nan | nan | nan | nan |
| Enemy | 99.0 | 100 | 99.5 | 73.1 | 87.6 | 100 | 93.4 | 72.1 | nan | nan | nan | nan |
| Player | 97.8 | 100 | 98.9 | 71.7 | 83.0 | 100 | 90.7 | 69.9 | nan | nan | nan | nan |
| PlayerScore | 100 | 100 | 100 | 100 | 97.0 | 94.2 | 95.6 | 97.4 | nan | nan | nan | nan |

## F.36    VENTURE DETAILS

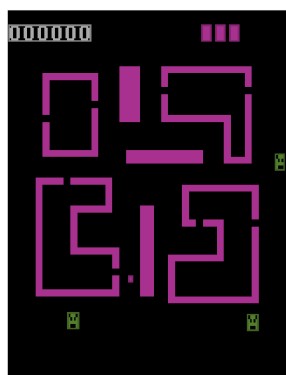

Your goal is to capture the treasure in every chamber of the dungeon while eliminating the monsters.

Table 42: Per class IOU on Venture

|  | Random | | | | DQN | | | | C51 | | | |
| --- | --- | --- | --- | --- | --- | --- | --- | --- | --- | --- | --- | --- |
|  | Pr | Rec | F-sc | IOU | Pr | Rec | F-sc | IOU | Pr | Rec | F-sc | IOU |
| Life | 100 | 100 | 100 | 100 | 100 | 100 | 100 | 100 | nan | nan | nan | nan |
| Hallmonsters | 46.2 | 99.6 | 63.2 | 84.1 | 26.7 | 98.3 | 42.0 | 84.6 | nan | nan | nan | nan |
| Player | 93.0 | 99.6 | 96.2 | 99.3 | 54.0 | 99.6 | 70.0 | 98.4 | nan | nan | nan | nan |
| Score | 100 | 100 | 100 | 100 | 100 | 100 | 100 | 100 | nan | nan | nan | nan |
| Goblin | 50.0 | 100 | 66.7 | 100 | nan | nan | nan | nan | nan | nan | nan | nan |
| Shot | 50.0 | 100 | 66.7 | 50.0 | 66.7 | 100 | 80.0 | 100 | nan | nan | nan | nan |
| Yellow_Collectable | 50.0 | 100 | 66.7 | 100 | nan | nan | nan | nan | nan | nan | nan | nan |
| Skeleton | nan | nan | nan | nan | 44.4 | 100 | 61.5 | 100 | nan | nan | nan | nan |
| Purple_Collectable | nan | nan | nan | nan | 66.7 | 100 | 80.0 | 100 | nan | nan | nan | nan |

## F.37    VIDEOPINBALL DETAILS

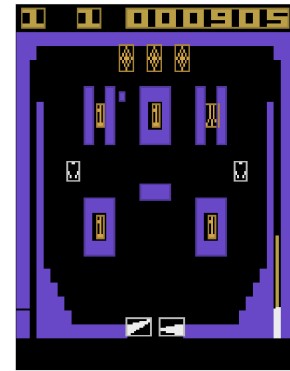

Your goal is to keep the ball in play as long as possible and to score as many points as possible.

Table 43: Per class IOU on VideoPinball

|  | Random | | | | DQN | | | | C51 | | | |
|  | Pr | Rec | F-sc | IOU | Pr | Rec | F-sc | IOU | Pr | Rec | F-sc | IOU |
|---|---|---|---|---|---|---|---|---|---|---|---|---|
| Score | 84.8 | 75.8 | 80.1 | 91.1 | 97.2 | 95.3 | 96.2 | 99.0 | nan | nan | nan | nan |
| DropTarget | 97.8 | 86.5 | 91.8 | 94.2 | 99.6 | 88.6 | 93.8 | 88.3 | nan | nan | nan | nan |
| LifeUsed | 98.0 | 100 | 99.0 | 100 | 99.6 | 100 | 99.8 | 100 | nan | nan | nan | nan |
| DifficultyLevel | 98.0 | 100 | 99.0 | 100 | 99.6 | 100 | 99.8 | 100 | nan | nan | nan | nan |
| Spinner | 98.0 | 99.7 | 98.8 | 76.1 | 99.6 | 100 | 99.8 | 76.5 | nan | nan | nan | nan |
| Flipper | 98.5 | 99.4 | 98.9 | 74.2 | 99.6 | 100 | 99.8 | 72.7 | nan | nan | nan | nan |
| Ball | 90.0 | 100 | 94.7 | 99.9 | 98.6 | 100 | 99.3 | 100 | nan | nan | nan | nan |
| Bumper | 98.0 | 100 | 99.0 | 99.6 | 99.6 | 100 | 99.8 | 99.9 | nan | nan | nan | nan |

## F.38 YARSREVENGE DETAILS

The objective is to break a path through the shield and destroy the Qotile with a blast from the Zorlon Cannon.

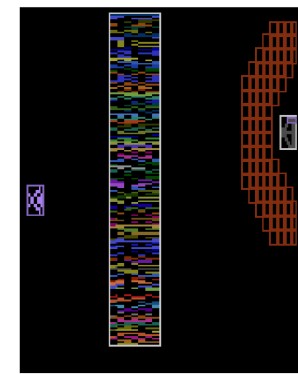

Table 44: Per class IOU on YarsRevenge

|  | Random | | | | DQN | | | | C51 | | | |
|  | Pr | Rec | F-sc | IOU | Pr | Rec | F-sc | IOU | Pr | Rec | F-sc | IOU |
|---|---|---|---|---|---|---|---|---|---|---|---|---|
| Player | 89.1 | 96.5 | 92.7 | 88.2 | 50.0 | 100 | 66.7 | 46.7 | nan | nan | nan | nan |
| Barrier | 40.9 | 94.1 | 57.1 | 100 | 50.0 | 100 | 66.7 | 100 | nan | nan | nan | nan |
| Shield_Block | 45.5 | 94.3 | 61.4 | 82.0 | 21.8 | 100 | 35.8 | 77.8 | nan | nan | nan | nan |
| Enemy | 98.2 | 88.7 | 93.2 | 98.6 | 100 | 100 | 100 | 100 | nan | nan | nan | nan |
| Enemy_Missile | 34.8 | 99.4 | 51.5 | 93.3 | 0.0 | nan | 0.0 | nan | nan | nan | nan | nan |
| Swirl | 65.0 | 92.9 | 76.5 | 63.7 | nan | nan | nan | nan | nan | nan | nan | nan |

## G    COMMON MISTAKES IN EXTRACTING AND DETECTING OBJECTS

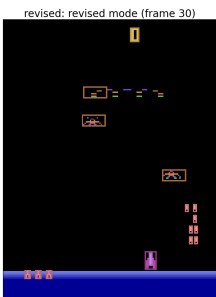 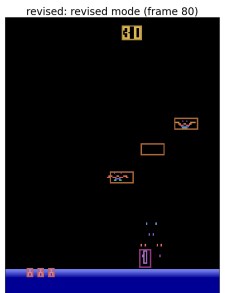 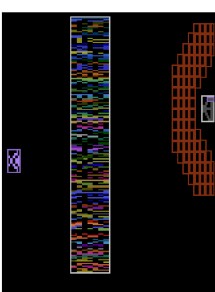

Figure 7: Animation and errors in the game of DemonAttack and YarsRevenge. We can see multiple particle effects and invisible objects. In the left we see the spawn animation of an enemy, i n the second image we see the death animation of the player and in the last we see the invisible shields in YarsRevenge. In all cases the objects are already detected even if it is not yet or not anymore visible to the player.

In this section, we will briefly discuss 2 common errors that can occur during detection and extraction based on the games DemonAttack and YarsRevenge.

**Case 1: Particle effects.** As described in Section 2, we primarily use positional information and the change of colors to identify objects in the visual detection of objects (VEM). It can happen that particle effects are incorrectly identified as objects, see Figure 7. In our RAM extraction we have defined the number and types of objects before extraction and concentrate on all game elements that are relevant for the game. Since these particle effects have no effect on the game, we deliberately do not detect them, which leads to a higher errors in F1 and IOU.

**Case 2: Invisible objects.** If objects disappear or appear in a game, there are various ways to realize this. The most common and simplest method, which is also used in most games, is to initialize objects only when they appear and to clear the memory when objects disappear. However, some games, such as DemonAttack or YarsRevenge (Fig. 7) use a different method. Here the objects are only set to invisible when they disappear or already exist before the objects appear. As such, these objects are also found and tracked by our REM method at an early stage, even though they have not yet appeared, which leads to an increased error. In many games we have therefore tried to find binary information about which objects are active so that those that are not, are not detected. This helps to minimize the error and increase the scores, as you can see in the updates scores in DemonAttack.

## H  DIFFERENCE BETWEEN ATARIARI AND OCATARI

| Game | Objects (AtariARI) | Objects (OCAtari) |
|---|---|---|
| Asterix | Enemies, Player, Lives, Score, Missiles | Enemies, Player, Lives, Score, Missiles |
| Berzerk | Player, Missiles, Lives, **Killcount**, Level, **Evil Otto**, Enemies | **Logo**, Player, Missiles, Enemies, Score, **RoomCleared** |
| Bowling | Ball, Player, **FrameNumber**, Pins, Score | Pins, Player, PlayerScore, **PlayerRound**, **Player2Round**, Ball |
| Boxing | Player, Enemy, Scores, Clock | Enemy, Player, Scores, Clock, **Logo** |
| Breakout | Ball, Player, Blocks, Score | Player, Blocks, **Live**, Score, Ball |
| Freeway | Player, Score, Cars | Player, Score, Cars, **Chicken** |
| Frostbite | Ice blocks, Lives, Igloo, Enemies, Player, Score | Ice blocks Blue, Ice blocks White, Score, Player Lives, Igloo, Enemies |
| Montezumas R. | **RoomNr**, Player, Skull, Monster, Level, Lives, **ItemsInInventory**, **RoomState**, Score | Player, Lives, Skull, Barrier, Key, Score, **Rope** |
| MsPacman | Enemies, Player, Fruits, **GhostsCount**, **DotsEaten**, Score, Lives | Lives, Score, Player, Enemies, Fruits |
| Pong | Player, Enemy, Ball, Scores | Player, Enemy, Ball, Scores |
| PrivateEye | Player, RoomNr, Clock, Score, Dove | |
| Q*Bert | Player, PlayerColumn, Red Enemy, Green Enemy, Score, TileColors | Cubes/Tiles, Score, Lives, **Disks**, Player, Sam, **PurpleBall**, **Coily**, GreenBall |
| Riverraid | Player, Missile, FuelMeter | **Score**, FuelMeter, **Tanker**, Lives, Player, **Helicopter**, Missile, **Bridge**, **Jet** |
| Seaquest | Enemy, Player, EnemyMissile, PlayerMissile, Score, Lives, DiversCount | Player, Lives, **OxygenBar**, Score, **Divers**, PlayerMissile, Enemy, EnemyMissile, DiverCount |
| SpaceInvaders | **InvadersCount**, Score, Lives, Player, Enemies, Missiles | Score, Lives, Player, Enemies, Missiles, **Satellite**, **Shield** |
| Tennis | Enemy, Scores, Ball, Player | Enemy, Scores, Ball, **BallShadow**, Player, Logo |

Table 45: All games, supported by both AtariARI and OCAtari with their respective object lists. Note that OCAtari returns a list of (x,y,w,h) per object and AtariARI provides the value written at a specific RAM position (x and y positions or the direct value, *e.g.* , scores and so on)

# I INSUFFICIENT INFORMATION IN ATARIARI

| Game | Reason |
|------|--------|
| Battlezone[1] | Unfinished |
| DemonAttack | Not all Demons are spotted |
| Hero | Missing Enemies |
| Q*Bert | Some Enemies, like Coily (Snake) are missing |
| Skiing[1] | Unfinished |
| RiverRaid | Important Elements (see above) are missing |
| Seaquest | Oxygenbar, Divers are missing |
| SpaceInvaders | Shields are missing |

Table 46: In Table 1 some games are marked with a ∼ to show that the RAM information provided by AtariARI are insufficient. This table gives a short reason while we marked each game.

# J REM VS VEM: SPEED PERFORMANCE

The following graph shows that we the RAM Extraction Method of OCAtari is, in average, $50\times$ computationally more efficient than the Vision Extraction method.

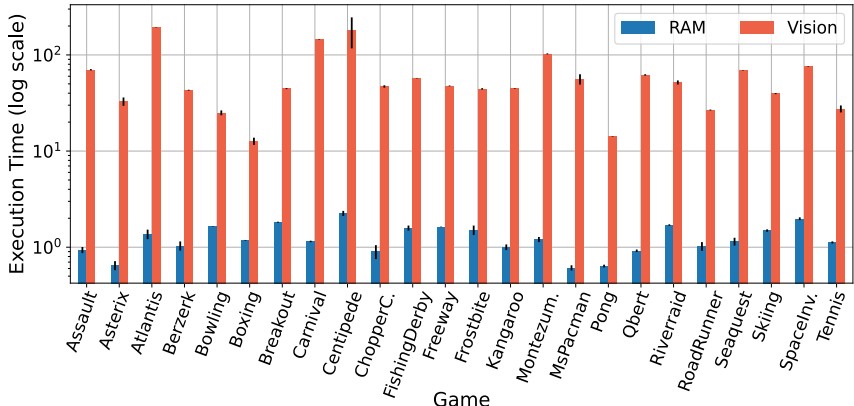

Figure 8: **Using the RAM extraction procedures leads to $50\times$ faster environments.** The average time needed to perform $10^4$ steps in each OCAtari game, using RAM extraction (REM), and our vision extraction (VEM).

---

[1]The games appear in the Github for AtariARI, but not in the associated publication (Anand et al., 2019). Also, the information does not seem sufficient to play with them alone so we did not indicate these games in Table 1 at all.

