# OpenReview forum: "OCAtari: Object-Centric Atari 2600 Reinforcement Learning Environments"
_ICLR.cc/2024/Conference — Submitted to ICLR 2024_

### Official Review · Reviewer_7uXz · 2023-10-27

**Soundness:** 3 good
**Presentation:** 3 good
**Contribution:** 2 fair
**Rating:** 5
**Confidence:** 4

**Summary:**

This paper presents a new benchmark, OCAtari, based on the ATARI 2600 RL environments. Instead of providing raw pixel observations to an RL agent, this benchmark allows the use of object-centric representations as input for an RL algorithm. The authors propose two methods to extract objects: one is a vision-based method that extracts bounding boxes by searching for a particular set of pixels per game, and the second is a method extracting the object coordinates from the game emulator RAM. In addition, the authors also generated a dataset of frames together with their detections, which can be used to evaluate the accuracy of an object-detection method on ATARI. Finally, OCAtari also allows to adjust the RAM state in order to generate particular or novel game situations, which could be used to generate new variants of existing Atari games.

**Strengths:**

- Standardised benchmarks are a cornerstone to advance research, as it allows to evaluate and compare different approaches on a level playing field. Being built on top of the well-known Atari benchmark further enables to compare against a wealth of related work on pixel-based RL.

**Weaknesses:**

- I think one of the main outstanding challenges in object-centric RL is how an agent can learn which are the particular objects in a game, and more importantly which are the relevant objects in a game. By introducing a predefined object extraction pipeline you basically fix an important part of an object-centric RL algorithm, and it's not said that the proposed object representation (i.e. coordinates and bounding box) is actually the best representation for an RL algorithm. So whereas this might be an interesting tool for developing and debugging object-centric methods, I'm in doubt of the value of this as a benchmark per se.

- In Figure 5, the pixel-based Deep PPO agents actually seem to outperform the object-centric ones in sample efficiency on some of the games (i.e. Asterix, Boxing and Freeway). This makes me wonder whether the current object-centric representation is actually fit for purpose. Of course the current policy that ingests the object-centric representation is pretty naive so that could be the point of the benchmark to further improve this.

**Questions:**

- For some games the object detection method seems to be far from perfect (i.e. DemonAttack, ChopperC.), or have a low IoU (i.e. IceHockey). It might be interesting to see what's going on in these environments in the qualitative results and/or appendix.

- If the environment gives you a list of objects, how does it handle "object permanence" over different frames? For instance, is the object at index i at time t the same object at index i at time t+1, or are the objects just put in random order in a list each timestep. Also, is the list of fixed size for a particular game, or does the list grow/shrink with the number of visible objects in a frame?

---

> ### Author Response · Authors · 2023-11-19
>
> Dear Reviewer,
>
> we thank you for your valuable feedback. We really appreciate all the time you have already spent to help us improve our paper. The points raised will help us to improve. We appreciate your indication that OCAtari greatly impacts the community that develops the object-centric RL algorithms. Let us now shortly address the opportunities.
>
> **Opportunity for improvements:**
>
> 1. **Object-centric states of OC-Atari might not be the most optimal.** OC-Atari can allow to separately train the object detection from the policy search, as we explain in the introduction, and does not prevent to train end-to-end agents. For our tested games, the representation extracted by REM allow matching deep agents’ performances. We cannot exclude that other object representations might be better suited for some tasks, we think that our open source code allows RL practitioners to convert or augment our object centric representations.
> 2. **Why are OCAtari agents learning slower than deep ones?.** We asked ourselves the same question. This is very likely due to the fact that we have done any hyperparameter tuning. We used the same hyperparameters as the original (deep) PPO agents, but the backpropagated gradients are much lower. Changing them on Boxing already leads to agents learning faster. Our goal is here to show that learning is feasible, we leave optimization of the learning speed for future work. We added this comment in the experiment section.
>
> **Raised Questions: \
> (1) For some games, the object detection method seems to be far from perfect.**
>
> We are constantly improving performance and adding new games. However, there are elements in some games that are captured by REM, but not by our VEM method, or vice versa, but do not participate in the gameplay (e.g. detecting particle objects). For example, we added to the appendix DemonAttack, in which the enemies spawn with a particle effect. This noise is not captured by REM, but is by VEM.
>
> We have noticed that some games in our table were still missing in the appendix, like the mentioned DemonAttack, ChopperCommand or IceHockey. We have now added details of all the games that are supported by REM in our appendix. We also added a small section about common mistakes which occur in games like DemonAttack, also in the appendix.
>
> **(2) If the environment gives you a list of objects, how does it handle "object permanence" over different frames?**
>
> In contrast to a purely visual approach where object permanence must be treated separately, REM detects blinking and overlapped objects. This can be used to train methods that are concerned with object permanence, such as [1]. We improved our discussion on this in the new manuscript version.
>
> [1] Liu, I. J., Ren, Z., Yeh, R. A., & Schwing, A. G. (2021, September). Semantic tracklets: An object-centric representation for visual multi-agent reinforcement learning. In _2021 IEEE/RSJ International Conference on Intelligent Robots and Systems (IROS)_ (pp. 5603-5610). IEEE.

---

> > ### Comment · Reviewer_7uXz · 2023-11-22
> >
> > I want to thank the authors for their comments. However, also considering the other reviewer's comments, I won't be increasing my score as 1) this still feels like an ongoing project, and this is a preliminary write-up and 2) I'd expect at least to see some indication in the experiments that object-centric methods can outperform the image-based ones.
> >
> > That being said, I think it's an interesting project, and maybe the authors should consider a submission to a more focused venue, for example, the Datasets and Benchmarks Track at Neurips.

---

### Official Review · Reviewer_BhyG · 2023-10-30

**Soundness:** 3 good
**Presentation:** 3 good
**Contribution:** 2 fair
**Rating:** 5
**Confidence:** 4

**Summary:**

This paper provides a new experimental platform based on the Atari 2600 to investigate object-centric representation in reinforcement learning. To extract object-centric representation, the authors implement two methods. One is the Vision Extraction Method (VEM) using computer vision techniques, and the other is the RAM Extraction Method (REM) based on AtariARI (Anand et al., 2019). Currently, VEM and REM cover 32 and 25 games, respectively. The authors also claim that OCAtari can change game elements or behavior by manipulating the RAM.

**Strengths:**

1. Creating the object-centric Atari Learning Environment greatly impacts the community that develops the object-centric RL algorithms.
2. The source code is available.

**Weaknesses:**

1. Although the introduction is well-written, it is still unclear why the ALE was selected. Please see my first question below.

**Questions:**

1. I agree that learning object-centric representation is one of the important research directions, but I am unsure whether the ALE is a good testbed because of its simplicity. In addition, there are several simulators that consider object-centric representation, such as VirtualHome (Puig et al., 2018), iGibson (Li et al., 2021), and AI2-THOR (Kolve et al., 2022). Would you explain why the ALE is selected in detail?
- Puig et al. (2018). VirtualHome: Simulating Household Activities via Programs. Proc. of CVPR.
- Li et al. (2021). iGibson 2.0: Object-Centric Simulation for Robot Learning of Everyday Household Tasks. Proc. of CoRL.
- E. Kolve et al. (2022). AI2-THOR: An Interactive 3D Environment for Visual AI. arXiv.
2. Recently, Aitchison et al. (2023) proposed a principled way to pick up a small subset of games. Their method makes it possible to reduce the computational cost because the algorithms are not evaluated on all the games. For example, they found that five games are enough. Is it possible to incorporate their method into OCAtari?
- M. Aitchison et al. (2023). Atari-5: Distilling the Arcade Learning Environment down to Five Games. Proc. of ICML.
3. What does the red circle in Figure 4 represent? In addition, the bounding boxes near the enemy are empty in MsPacman. It suggests that OCAtari does not treat tiny rectangles (foods) as objects. Is my understanding correct?
4. Figure 5 seems interesting. I would like to know why the pixel-based PPO agent learns slightly faster than the object-centric PPO agent. Specifically, I expected the OC-PPO agent to learn faster because it has good state representation, but the result is the opposite. Would you discuss this point in detail?

---

> ### Author Response · Authors · 2023-11-19
> **We appreciate your feedback, please consider our clarifications and updated version.**
>
> Dear Reviewer,
>
> we thank you for the valuable feedback. We really appreciate all the time you have already spent to help us improve our paper. The points raised will help us to improve. We are happy that you think that OCAtari greatly impacts the community that develops the object-centric RL algorithms. Let us now shortly address the opportunities.
>
> Opportunities for Improvement:
>
>
>
> 1. **Why selecting Atari games ?**  Thank you for this. We have answered this concern in the general comment. Shortly: \
> * ALE is the most used benchmark for existing RL algorithm,  \
> * ALE’s games diversity covers many learning aspects and challenges  \
> * The lack of human reasoning capabilities of ALE trained agents has yet to be addressed.  \
>
> 2. **Incorporating Aitchison et al. (2023) in OCAtari.** Thank you very much for this amazing resource. We will concentrate our next efforts to cover the 3 remaining games that are recommended (and will of course cite this work). We are nearly done with BattleZone already. We also believe that many other games (such as Skiing for difficult credit assignment, and e.g. Montezuma’s Revenge or Pitfall for sparse reward) provide challenges that are worth exploring as well. We added this comment on page 8. \
>
> 3. **Red circles in Figure 4.** Good point. These circles represent are supposed to show different cases where the REM and VEM differ, as the objects are present in the environment but blinking (in MsPacman), exploded (SpaceInvaders) or partially occluded (FishingDerby). We added this to the caption of the figure (and removed the one from Skiing).  \
>
> 4. **Why are OCAtari agents learning slower than deep ones?** We asked ourselves the same question. This is very likely due to the fact that we have done any hyperparameter tuning. We used the same hyperparameters as the original (deep) PPO agents, but the backpropagated gradients are much lower. Changing them on Boxing already leads to agents learning faster. Our goal is here to show that learning is feasible, we leave optimization of the learning speed for future work. We added this comment in the manuscript.

---

### Official Review · Reviewer_wGg4 · 2023-11-01

**Soundness:** 3 good
**Presentation:** 3 good
**Contribution:** 2 fair
**Rating:** 5
**Confidence:** 4

**Summary:**

The paper presents a modification of the Atari-game Arcade Learning Environment (ALE) that augments the traditional pixel observations with information about objects and their bounding boxes. The new library, OCAtari, uses two different object extraction methods, one vision based, the other RAM-based. The paper describes the relative strengths and weaknesses of these two approaches, and demonstrates that the object-centric representations can be used for reinforcement learning.

**Strengths:**

The library can serve as a best-case result for comparing against approaches that do Atari object detection without access to the RAM state. It also allows research to proceed on designing agents that exploit object-centric representations without first waiting for object-detection methods. The paper also argues that when visual features are not needed, this library can dramatically speed up the training process by eliminating much of the rendering pipeline.

**Weaknesses:**

1. I am somewhat skeptical that this library will be broadly useful to the field. The paper presents a publication-count-based argument that there is a need for object-centric Atari environments. I don't draw the same conclusion from the presented evidence. The authors may be surprised to learn that one of the earliest Atari + reinforcement learning papers (from 2008!) dealt with this exact question: Diuk, Cohen & Littman's paper on Object-Oriented MDPs. The fact that Atari did not become a popular testing ground for reinforcement learning until later, with the advent of the ALE, suggests that the lack of well-defined object-centric representations was not what was holding back adoption, but rather the availability of a wide range of domains featuring a common interface.

2. I am also skeptical of the authors' prediction that they will "complete what we have started" and add the remaining ALE games. What would it mean to "complete" this project? It seems like there are substantial hurdles left to overcome in the remaining games, particularly since without Atari ARI as a guide, they will require much more reverse engineering to extract objects from the RAM.

3. The paper claims that the library allows new modifications of existing games. They present "hard-fire Pong" as an example, but this seems to already be possible under the existing ALE. I suspect many of the proposed variations are already possible as well, since the ALE already provides `set_ram` functionality.

4. I am unconvinced by the argument that the REM object detection will amount to a training-time speedup. For example, in Ms. Pacman, the object detection does not work for walls, so visual observations are necessary even when objects are available. (Incidentally, the paper claims that the walls in Ms. Pacman are static, but this is incorrect; they change after a certain number of completed levels.)

**Questions:**

My main question for the authors is: if objects are indeed so important, why not build an environment (or improve an existing one) that supports objects natively, rather than reverse-engineering one from the ALE? What's so special about Atari?

---

> ### Author Response · Authors · 2023-11-19
> **Thank you for your feedback. Please consider our answer:**
>
> Dear Reviewer,
>
> we thank you for the valuable feedback. The points raised have help us to greatly improve the manuscript. Thank you for your positive comments on the use of our library to train object-centric RL agents. Let us address the different points that you have raised.
>
> Opportunities for Improvement:
>
>
>
> 1. **The lack of object-centric representations has not held back the adoption of ALE.** ALE has indeed made available a wide range of domains featuring a common interface, with resources requirements, making ALE the go-to framework for testing RL methods. The observed trend in the emergence of object-centric and neurosymbolic methods is notable, as reflected in Fig.1. OCAtari provides benefits for a wide range of domains. We added a paragraph in the introduction to strengthen this claim. \
>
> 2. **Authors might not continue the development to cover more games.** Since the paper submission, our coverage has expanded to include 7 additional games (compared to our original manuscript), with enhanced detection performance for two of them. In our provided codebase, the "_scripts_" folder contains various tools for reverse engineering games. Notably, our coverage surpasses that of AtariARI, now spanning more than twice as many games. As you can see, we plan to provide as many games as possible and do not require guidance by AtariARI to do so.  \
>
> 3. **Games are already modifiable within ALE**. Yes they are. However, to modify the behavior of the games, one first needs to understand how the information is processed and where it is stored. Our Open Source OCAtari framework makes this information process transparent. For example, to understand how to remove particular types of enemies, one first needs to know the RAM positions encoding the enemy type, but also what values correspond to these enemies, explicitly provided by OCAtari. We made this point clearer in the manuscript in section 2.3. \
>
> 4. **REM might not save time.** As you said, for very few games, such as MsPacman, the background is not encoded in our object-centric description. This is why, for such games, hybrid approaches might be necessary, as described in our limitations section. But even hybrid approaches could benefit from turning off expensive object detection modules, and thus save many resources.  \
> The background changes after some levels, but that still constitutes a fixed number of backgrounds, these are not procedurally generated (thus equivalent to a problem of having one). Extracting a maze representation to use e.g. the A* algorithm, together with an object-centric RL trained algorithm might be an interesting approach that also would benefit from OCAtaris object extractions. We expanded our future work section to now include this discussion.
>
> Question:
>
> 1. **Why not creating directly object-centric environments ?** In other works, we had created more modern environments for the development of different neurosymbolic approaches, but have been many times asked to compare to deep approaches on ALE games. Beyond having been adopted by most deep RL practitioners, ALE provides a wide range of tasks featuring a common interface, with many RL specific caveats available (e.g. sparse reward, difficult credit assignment, …etc). These environments have been developed to challenge humans, and not AI, and thus don’t include any experimenter's bias. They are also cheap to run compared to more modern games. We added a paragraph to strengthen our motivation about using Atari in the introduction.

---

> > ### Comment · Reviewer_wGg4 · 2023-11-22
> > **Summary of Review**
> >
> > Overall, my feelings about this work are roughly the same. I'm not sure whether it will be valuable to the field in its current form, but it seems like it is almost there.
> >
> > For me, the main benefit lies in offering a side-by-side comparison of object-oriented vs. pixel-based representations for a widely popular domain. I don't find the arguments about training time speed-up or modifications to existing games convincing.
> >
> > What I would prefer is that the authors focus on this side-by-side comparison aspect, and use their library to survey several object-oriented approaches and several pixel-based approaches, and kick off a discussion about whether or not object-oriented indeed helps as much as proponents of that approach seem to think.
> >
> > It might be that they do, in which case, we should try to import object-oriented techniques into our pixel-based models. It also might be that they do not (as Fig 5 seems to suggest), which would imply either that pixel-based methods are already aware of objects, or that object-awareness is not as important as we might think. Helping understand these trade-offs is where I think this library will have the most value.

---

### Official Review · Reviewer_Lfyp · 2023-11-07

**Soundness:** 3 good
**Presentation:** 2 fair
**Contribution:** 2 fair
**Rating:** 3
**Confidence:** 4

**Summary:**

The authors consider an interesting gap in the RL benchmarks i.e. the evaluation and training of object centric approaches. To this end, this paper introduces Object-Centric Atari, a set of environments that provides object-centric representations of ALE.  They show that OCAtari can be used to train or evaluate any part of an object-centric RL agent-- including object detection methods that extract objects and  extract categories from the extracted embeddings or be used in supervised learning directly. They build on top of AtariARI.

**Strengths:**

This work provides a concrete benchmark/dataset to train and evaluate object centric properties of RL approaches. Key strengths of the paper include:
*    The benchmark can serve a nice tool to test the abstraction ability of our methods today as humans seamlessly can detect objects and reason in the space of object oriented.
*    It can inform the quality of representations learned in ALE domains and further if the methods have abilities such as compositional generalization, etc.
*    The work is valuable to provide the open-source implementation of the benchmark.
*    Finally, the authors also provide documentation to customize these environments etc.

**Weaknesses:**

The paper offers interesting contribution, however I believe the paper is not up to the mark for the ICLR conference venue. I find the following issues major limiting factor in recommending acceptance:
*   The topic and contribution is relevant, however it is unclear to me immediately what this buys us for methods not focused at object central. For e.g. a method might be able to achieve very good performance but not do well on object centric evaluation.
*    What is missing and would be nice to see baselines of representation learning methods to showcase the benchmark's utility further?
*    The contributions on top of AtariARI seem not substantially different, for e.g. can we not access VEM provided information.

**Questions:**

I would be happy to reconsider my score and engage during discussion period with the following questions:

*    What is the contribution here from the learning perspective? May be I am missing something here, but is it correct to understand that the primary contribution lies in a wrapper around the ALE benchmark to be able to provide an object centric evaluation for RL agents in a way that combines REM (from prior work) with previously established vision modules to annotate objects?
*    Next, the paper shows that the current methods do not necessarily have a great understanding of objects? Is that strictly necessary to solve the tasks? While I value object centric research, I am unsure if the agents would strictly needs that to solve tasks and should be evaluated on this ability more strictly for methods who do not bake such an inductive bias in the approaches?
*   "To extract objects, OCAtari uses either a Vision Extraction Method (VEM) or a RAM Extraction Method (REM), that are depicted in Figure 2." What were the specific techniques used to perform vision. I was unable to find this information in the manuscript.
*    The authors mention that RL methods are hard to evaluate due to the non-determinism of the approaches -- how does OCAtari overcome this problem?

---

> ### Author Response · Authors · 2023-11-19
> **Thank you for your work. Let us address your concerns.**
>
> Dear Reviewer,
>
> we really want to thank you for all the effort you have put in the reviews, and the valuable feedback you have provided us. We agree that OCAtari can serve a nice tool to test the abstraction ability of RL methods in their ability to produce human-like object-centric reasoning. Your raised points have greatly helped us to improve our manuscript.
>
> Let us address the opportunities (which we enumerated to make the assignment easier).
>
> Opportunities for Improvement:
>
> 1. **What benefit for classic deep methods?** The OCAtari framework, serves as an important asset for advancing object-centric reinforcement learning (RL) and facilitating comparisons with established deep RL methodologies. OCAtari extends its utility to conventional deep RL approaches, as it can e.g. be employed together with Language Model (LLM) that guide agents training using on game manuals’ instructions (and object grounding) [1, 2]. Moreover, OCAtari's reveals games’ inner workings, allowing the development of modified game versions for the evaluation of e.g. continual RL methods, as done in [3] on Doom. \
>
> 2. **Baselines of representation learning methods.** We provide comparison in detection capabilities of the two SOTA methods evaluated on Atari: SPACE [4] and MOC [5] in table 3. These methods obtain an average few shot classifier that goes up to 22% for SPACE and 84% for MOC. OCAtari directly provides the object classes, that can be used to train the classifiers. In our F-Score computations, we only consider that an object is correctly detected if it has the correct class. We added this comment in our manuscript in section 3.1
> 3. **Not much contribution over AtariARI.** AtariARI only indicates the location of raw information within the RAM, whereas our framework provides reconstructed object-centric information. This involves intricate computations, such as deriving the objects’ positions from anchors and offsets, looking up potential presence indicators (e.g. for objects that have been destroyed), ensuring that our models genuinely acquire object-centric state descriptions. Moreover, our coverage spans 35 games, a substantial increase compared to AtariARI's 16, and continues to expand. We rephrased section 3.3 to better emphasize this.
>
> Questions:
>
>
>
> 1. **What is the learning contribution?** Our key contribution is the introduction of the OCAtari environments. We showcased the feasibility of training PPO agents that use a 2 layer MLP on the object-centric state descriptions across different games. Our efficient ram extraction method (REM) stands out for getting object-centric information from RAM values, surpassing the energy efficiency of neural networks or hard-coded vision detection algorithms like VEM. The development of efficient or reliable learning algorithms is not the focus of this paper. We are providing a resource-efficient alternative for developing, evaluating and training object-centric RL *reasoning* agents, that can be based on e.g. logic, as done in [6, 7, 8].  \
>
> 2. **Is object-centricity necessary to solve a task?** While object-centricity (OC) is not inherently indispensable for solving RL games, numerous studies in RL research highlight its importance in elucidating agent reasoning, prevent misalignment, and potentially correct them. Notably, studies such as [1, 2] leverage Language Model-based Learning (LLMs) to realign agents, relying on natural language descriptions of environment goals for this purpose with great success. This underscores why we need to enhance our understanding of agent behavior and ensuring their proper alignment with the intended objectives. According to these studies, OC is a necessary step to achieve it. We added this in our update of the introduction.
> 3. **How is VEM working?** We have improved the the ***VEM: the Vision Extraction Method*** paragraph in our manuscript, page 3: “At each stage, objects are extracted using color-based filtering and an object position priority. For example, Pong consists of 3 primary objects and 2 HUD objects, each assigned constant RGB values (cf. Figure 2). Contrary to the paddles, the HUD scores always appear above the white threshold, which allow us to distinguish these two same-colored objects. This technique allows to reliably extract the depicted objects.”
> 4. **How does OCAtari overcome evaluation on non-deterministic approaches?** All our OCAtari environments are compatible with the *Deterministic*, *NoFrameskip* and *v0* to *v5* variations of gymnasium. This helps to reproduce experiments and make it easier to provide benchmarks and datasets. We added this information into our experimental details section in the appendix. Thank you for raising this up, we add this comment to our manuscript. But you are right, the non-determinism is not the important aspect of OCAtari, we modified the paper to just express that RL benchmarks are important.

---

> > ### Author Response · Authors · 2023-11-19
> > **References**
> >
> > [1] Zhong, V., Hanjie, A. W., Wang, S., Narasimhan, K., & Zettlemoyer, L. (2021). Silg: The multi-domain symbolic interactive language grounding benchmark. _Advances in Neural Information Processing Systems_, _34_.
> >
> > [2] Wu, Y., Fan, Y., Liang, P. P., Azaria, A., Li, Y., & Mitchell, T. M. (2023). Read and reap the rewards: Learning to play atari with the help of instruction manuals. _Advances in Neural Information Processing Systems_, _36._
> >
> > [3] Tomilin, T., Fang, M., Zhang, Y., & Pechenizkiy, M. (2023). COOM: A Game Benchmark for Continual Reinforcement Learning.  _Advances in Neural Information Processing Systems_, _36._ \
> >  \
> > [4] Lin, Z., Wu, Y. F., Peri, S. V., Sun, W., Singh, G., Deng, F., ... & Ahn, S. (2020). Space: Unsupervised object-oriented scene representation via spatial attention and decomposition.
> >
> > [5] Delfosse, Q., Stammer, W., Rothenbächer, T., Vittal, D., & Kersting, K. (2023, September). Boosting object representation learning via motion and object continuity. In Joint European Conference on Machine Learning and Knowledge Discovery in Databases
> >
> > [6] Goodall, A. W., & Belardinelli, F. (2023). Approximate Shielding of Atari Agents for Safe Exploration. _arXiv preprint arXiv:2304.11104_.
> >
> > [7] Delfosse, Q., Shindo, H., Dhami, D., & Kersting, K. (2023). Interpretable and Explainable Logical Policies via Neurally Guided Symbolic Abstraction. _arXiv preprint arXiv:2306.01439_.
> >
> > [8] Hasanbeig, H., Kroening, D., & Abate, A. (2023). Certified reinforcement learning with logic guidance. _Artificial Intelligence_, _322_, 103949.

---

### Author Response · Authors · 2023-11-19
**Thank you all, we improved our manuscript based on your comments.**

Dear reviewers,

We are immensely grateful for the feedback we got and thank all reviewers again. In this rejoinder, we address the most common concern about our inclination towards the Arcade Learning Environment (ALE) as a novel object centric framework.

Reviewers seem to agree that object-centric (OC) representation is an important research direction, but have concerned about the use of Atari games to conduct this research. Let us express why OCAtari games will help RL practitioners.

**ALE is the most used benchmark for existing RL algorithm.** As pointed out by Reviewer **7uXz**, the **“**well-known Atari benchmark further enables us to compare against a wealth of related work on pixel-based RL”. Bringing OC to ALE allows comparing and adapting the multitude of existing pixel-based approaches to more human reasoning tasks. As shown in table 1, this benchmark has been more used than the 8 next benchmarks altogether. ALE games provide low resolution frames, but allow tackling many RL challenges while avoiding unnecessary computational needs.

 \
**ALE’s games diversity covers many learning aspects and challenges.**
As already mentioned above, ALE games allow practitioners to tackle many RL challenges, such as difficult credit assignment (Skiing), sparse reward (Montezuma’s Revenge, Pitfall), partial observability [3], generalization [2], sample efficiency [5], environment modeling [6]. Most existing object-centric environments (such as those provided by **BhyG**) model robots solving domestic tasks or concern navigation systems. Such environments do not include the diversity of the ALE games.

**The lack of human reasoning capabilities of ALE trained agents has yet to be addressed.** Lake et al. [1] used Frostbite to demonstrate that deep agents can’t decompose the world into multi-steps sub-goals, such as getting some objects and avoiding others. Farebrother et al. [2] show that deep agent can’t generalize and easily transfer among similar Atari games, even in cases where only e.g. the background color is changed.

Wu et al. [0] solve the difficult credit assignment problem with a deep agent. They use LLM to guide a deep agent based on the game's manual that associate the instructions with object-centric states.

**OCAtari will also benefit pixel-based approaches.** The last example show how OCAtari can be used to help deep agents. LLMs could be further used to guide deep agents, to help with other problems such as reward sparsity. Finally, OCAtari making transparent how the information is stored in the RAM, it allows to easily create variations of ALE environments, that can help deep agents to generalize.

These arguments improve the motivations for the development of Object Centric Atari environments. Here is the list of the corresponding modifications in our paper. Thank you all. We: \




* Added a small paragraph, motivating the usage of the ALE to the introduction,
* Improved clarity on the fact that OCAtari is built upon the ALE and uses its features for e.g., the RAM manipulation in Section 2.3,
* Added specific use cases how OCAtari’s RAM manipulation could help pixel-based approaches in Section 2.3,
* Added a comparison to AI2-THOR and iGibson to the Related Work section,
* Added some games (see Tables),
* Added more detailed versions for the missing games in the App. F,
* Replacing Table 2 with heatmap to save space (Old Table 2 now in App. F),
* Added a section about particle effects and invisible objects in the App,
* Repaired DemonAttack.

---

> ### Author Response · Authors · 2023-11-19
> **References**
>
> [1] Lake, B. M., Ullman, T. D., Tenenbaum, J. B., & Gershman, S. J. (2017). Building machines that learn and think like people. _Behavioral and brain sciences_, _40_, e253.
>
> [2] Farebrother, Jesse, Marlos C. Machado, and Michael Bowling. "Generalization and regularization in dqn." _arXiv preprint arXiv:1810.00123_ (2018).
>
> [3] Hausknecht, M., & Stone, P. (2015, September). Deep recurrent q-learning for partially observable mdps. In _2015 aaai fall symposium series_.
>
> [4] Dabney, W., Rowland, M., Bellemare, M., & Munos, R. (2018, April). Distributional reinforcement learning with quantile regression. In _Proceedings of the AAAI Conference on Artificial Intelligence_ (Vol. 32, No. 1).
>
> [5] Espeholt, L., Soyer, H., Munos, R., Simonyan, K., Mnih, V., Ward, T., ... & Kavukcuoglu, K. (2018, July). Impala: Scalable distributed deep-rl with importance weighted actor-learner architectures. In _International conference on machine learning_ (pp. 1407-1416). PMLR.
>
> [6] Danijar Hafner, Timothy P. Lillicrap, Mohammad Norouzi, Jimmy Ba:
>
> Mastering Atari with Discrete World Models. ICLR 2021
>
> [7] Machado, M. C., Bellemare, M. G., Talvitie, E., Veness, J., Hausknecht, M., & Bowling, M. (2018). Revisiting the arcade learning environment: Evaluation protocols and open problems for general agents. _Journal of Artificial Intelligence Research_, _61_, 523-562.
>
> [8] [https://gymnasium.farama.org/environments/atari/complete_list/](https://gymnasium.farama.org/environments/atari/complete_list/)
>
> [9] Wu, Y., Fan, Y., Liang, P. P., Azaria, A., Li, Y., & Mitchell, T. M. (2023). Read and reap the rewards: Learning to play atari with the help of instruction manuals. Advances in Neural Information Processing Systems, 36.

---

### Meta-Review · Area_Chair_EKPw · 2023-12-05

**Metareview:**

This paper augments the Arcade Learning Environment (ALE) to provide information about objects and their bounding boxes, generating an “object-centric representation” of the ALE. Overall, it was agreed upon that the paper is not ready for publication. Many of the questions asked by the reviewers, if properly answered, have the potential to improve the paper and I recommend the authors to look at them. Two crucial points for this decision of recommending rejecting the paper were the fact that the paper actually seems to be a work in progress and that the potential impact/usefulness of OCAtari is not clear. Finally, I want to point out that Machado et al. (2018) did introduce game variations to Atari 2600 games by supporting different game modes and difficulties in the Arcade Learning Environment (ALE).

**Justification For Why Not Higher Score:**

As per the meta-review, none of the reviewers actually think the paper is ready for publication. While introducing a new benchmark is potentially exciting, the reviewers were left with the impression that this was still a work in progress (e.g., based on the number of games supported) and that the paper would greatly benefit from some additional results to motivate and justify the proposed benchmark.

**Justification For Why Not Lower Score:**

N/A

---

### Decision · Program_Chairs · 2024-01-16

Reject